# Relaxing Pointwise Imitation in Knowledge Distillation with Mean-Projection Matching

## Abstract

Logit-based knowledge distillation (KD) typically trains a student by matching the teacher's predictive distribution. This pointwise target transfers useful soft information, but can also pass along miscalibrated, biased, or wrong teacher predictions. We revisit the supervision object and propose Mean Projection (MP), a distillation objective on the square-root probability sphere. For each gold-label subset in a mini-batch, MP decomposes the empirical mean vector of teacher predictions into a direction and a length, where the length is the average of sample-level scalar projections onto that direction. MP uses two bounded matching terms: one aligns the subset-level mean direction, and the other aligns these sample-level projections. This preserves the subset summary without forcing the student to match the full teacher distribution pointwise; in the single-sample limit, each term reduces to the squared Hellinger distance. Empirically, a natural language inference (NLI) diagnostic motivates the imperfect-teacher setting. On fine-grained classification, MP gives clear gains on visually similar datasets, where it matches the strongest logit-distillation baseline on accuracy while showing an expected calibration error (ECE) advantage among distillation methods. Mechanism analysis shows that MP predicts the gold label on teacher-wrong samples 2.9 percentage points more often than KD and copies teacher mistakes 4.1 percentage points less often. Further analysis traces these effects primarily to the square-root geometry, with subset relaxation contributing additional inheritance gains.

## 1 Introduction

Logit-based knowledge distillation (KD) typically trains a student by matching the teacher's predictive distribution, often through a temperature-scaled Kullback-Leibler (KL) divergence (Hinton et al., 2015; Zhao et al., 2022; Sun et al., 2025). This form of supervision is attractive because it transfers information beyond the hard label, including the teacher's confidence on both the target and non-target classes. Other distillation lines instead transfer intermediate representations or attention maps (Romero et al., 2015; Yim et al., 2017; Zagoruyko & Komodakis, 2017; Heo et al., 2019; Tian et al., 2020; Chen et al., 2021; 2022; Guo et al., 2023), but sample-level output supervision remains a common and influential choice.

This pointwise object can become fragile when the teacher is imperfect. Trained networks can be miscalibrated, overconfident, or systematically biased (Li et al., 2017; Lukasik et al., 2022; Hamidi et al., 2024; Zhang et al., 2025; On et al., 2026). When such imperfections persist on the data, pointwise matching can pass them to the student. Existing remedies reweight or filter samples, rectify teacher outputs, learn residual corrections, replace the divergence, change the teacher's training objective, or combine multiple teachers (Lukasik et al., 2022; Guo et al., 2024; Hamidi et al., 2024; Zhang et al., 2025; Yang et al., 2025; On et al., 2026; Tybl & Neumann, 2026). These methods improve the use of teacher information, but they still often apply teacher predictions sample by sample. This leaves a basic question open: can logit-based KD change how it uses these predictions by relaxing exact pointwise copying?

This suggests a shift in viewpoint. Here, we ask how teacher predictions should be used to implement supervision. Useful teacher information may also be carried by summaries of these predictions, not only by the full predictive distribution for each individual sample.

We propose Mean Projection (MP), a simple objective that changes the supervision object under the logit-based setting. For each gold-label subset in a mini-batch, MP represents the teacher predictions on the square-root probability sphere by their empirical mean vector. This mean vector decomposes into a direction and a length. The length can in turn be expressed as the average of sample-level scalar projections onto that direction. This gives two matching terms: mean-direction matching between the teacher and student subsets, and sample-level projection matching along the teacher mean direction.

This form of supervision admits an admissible-set interpretation on the sphere: the student preserves teacher information along the mean direction while leaving orthogonal components less constrained. This view suggests a testable prediction in aggregate: if MP relaxes pointwise copying, the student should tend to recover more teacher-wrong samples and copy fewer teacher mistakes.

Our contributions are summarized as follows.

**(i) We reformulate the supervision object in logit-based KD.** We map each $K$-dimensional predictive distribution through a coordinatewise square-root transformation, placing it on the unit positive sphere. One-hot classes become orthogonal coordinate axes and decision regions are fixed by the sphere geometry. In this space, MP replaces a pointwise teacher target with a subset-level mean direction and a sample-level projection, giving an admissible-set interpretation of teacher supervision (Figure 1).

**(ii) We give a statistical interpretation and limiting case.** The empirical mean vector of a class-wise teacher subset decomposes into a direction and a length, with the length expressed as the average of sample-level projections onto that direction. MP follows this decomposition with two terms: mean matching captures subset-level direction, while projection matching captures sample-level coordinates along that direction. From a vMF-inspired directional-statistics perspective, the two terms correspond to empirical direction and concentration-related quantities (Section B.5). In the single-sample limit where each nonempty subset contains a single example, MP reduces to pointwise squared Hellinger matching.

**(iii) We evaluate MP through imperfect-teacher diagnostics and fine-grained classification.** A diagnostic NLI study shows the imperfect-teacher setting directly: the teacher can underperform a cross-entropy student, and MP performs better than other distillation baselines for the weaker student in the harder adversarial setting (Section 4.1). On fine-grained classification, MP shows an ECE advantage among distillation methods, with top-1 on par with the strongest baseline (Section 4.2). Section 4.4 provides additional analysis. We study robustness under Office-Home distribution shift using the $OOD-ID$ NLL gap, probe teacher non-target information with a DKD coefficient sweep, and analyze MP design choices through ablations and the single-sample Hellinger endpoint.

**(iv) We analyze teacher-error inheritance and its sources.** A per-sample mechanism analysis shows that MP-trained students recover teacher-wrong samples, i.e., predict the gold label when the teacher is wrong, 2.9 pp more often than KD-trained students, and mimic teacher mistakes, i.e., copy the teacher's wrong prediction, 4.1 pp less often (Section 4.3, Table 3). A further decomposition shows that these effects combine MP's square-root geometry and subset relaxation: the single-sample endpoint accounts for much of the calibration benefit and a substantial part of the change in recovery and mimicry, while the subset component adds further teacher-error inheritance gains (Section 4.4, Table 6).

## 2 Related Work

We position MP relative to two lines of work: logit-based knowledge distillation and broader machine learning objectives that replace exact point targets with summaries, anchors, or relations.

### 2.1 Knowledge distillation

KD methods differ in the object transferred from teacher to student. Logit-based distillation matches predictive distributions (Hinton et al., 2015); feature distillation transfers intermediate representations (Romero et al., 2015; Yim et al., 2017; Zagoruyko & Komodakis, 2017; Heo et al., 2019; Tian et al., 2020; Chen et al., 2021; 2022; Guo et al., 2023); and relation distillation transfers information such as sample-pair similarities

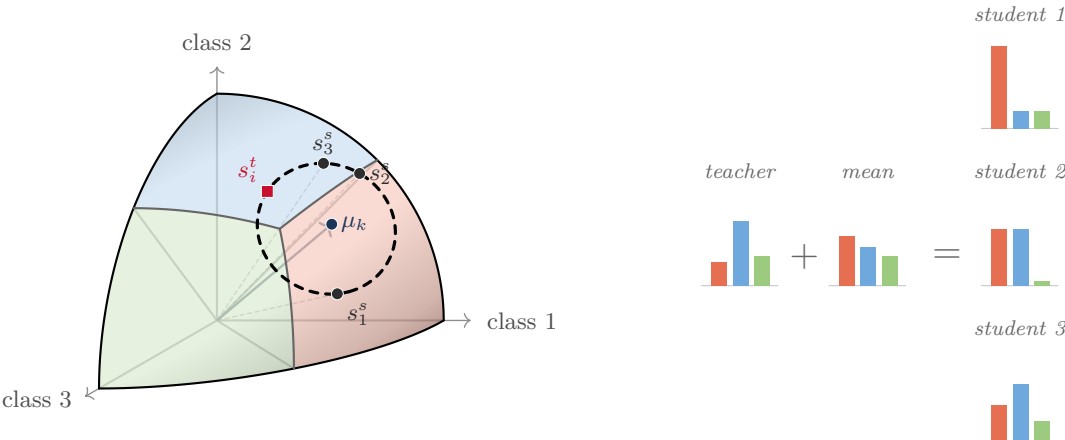

Figure 1: **Projection matching gives a conceptual admissible set.** *Left.* On the $K = 3$ unit positive sphere, the teacher subset mean direction $\mu_k^t$ defines a projection axis. A teacher prediction $s_i^t$ fixes the scalar value $(\mu_k^t)^\top s_i^t$, and all student predictions with the same value lie on the dashed arc. Conceptually, the teacher point is relaxed from a unique target into a projection level set. Some points on this set lie in the gold-label region, whereas others remain in a wrong-class region. *Right.* Histogram view of the same projection level with several possible student realizations. MP is used together with cross-entropy; see Section C for a detailed set description.

or structural relations (Park et al., 2019; Peng et al., 2019; Tung & Mori, 2019). MP operates only on teacher and student output distributions, so its direct competitors are logit-based distillation methods.

Recent logit-based methods mainly refine how the teacher distribution is weighted, decomposed, or corrected. DKD (Zhao et al., 2022) separates target-class and non-target-class terms; MLKD (Jin et al., 2023) augments instance-level KL with batch-level and class-level alignment terms based on teacher-student prediction correlation matrices; CTKD (Li et al., 2023) introduces a per-sample temperature; and LSKD (Sun et al., 2024) standardizes logits before the softmax to reduce logit-scale mismatch. Label-aware methods edit or mask the teacher target: LA (Wen et al., 2021) and LR (Lan et al., 2025) modify the teacher distribution when it conflicts with the gold label; RC (Cao et al., 2023) amplifies the gold-label probability in the student's output; and RLD (Sun et al., 2025) masks classes ranked above the true class and distils the remaining distribution. REDistill (Tybl & Neumann, 2026) replaces KL with a robust power divergence. Other work addresses teacher imperfection through reweighting, uncertainty modeling, filtering, residual learning, or multi-teacher combination (Li et al., 2017; Lukasik et al., 2022; Guo et al., 2024; Hamidi et al., 2024; Zhang et al., 2025; Yang et al., 2025; On et al., 2026).

MP takes a different route: it changes the object being matched, using a subset-level mean direction and a sample-level projection on the square-root probability sphere.

## 2.2 Beyond exact pointwise objectives

MP also connects to broader objectives that move beyond exact pointwise targets. In machine learning, such objectives often use distributional summaries, shared anchors, or relations among samples.

*Distributional summaries.* One line of work replaces pointwise targets with aggregate statistics of a set or distribution. Kernel mean embeddings and related divergences (Jenssen et al., 2006; Gretton et al., 2012; Muandet et al., 2017; Yu et al., 2024) compare distributions through summary statistics. Mean vector component analysis (Jenssen, 2013) is especially close in spirit: for nonnegative data, the empirical mean vector carries information through both its direction and its magnitude. MP brings this intuition to predictive distributions through the square-root transformation. The summary is a mean of square-root probability vectors, which can be more stable than individual teacher targets.

*Anchors, prototypes, and self-distillation centers.* Prototypical networks (Snell et al., 2017) represent each class by the mean of its support embeddings, while proxy-based metric learning uses learnable class anchors (Movshovitz-Attias et al., 2017; Kim et al., 2020). Related ideas also appear in self-supervised learning: SwAV (Caron et al., 2020) matches assignments to shared prototypes, and DINO (Caron et al., 2021) applies a softmax over the similarities between an embedding and learned prototypes, and uses centering and sharpening to stabilize teacher-student alignment; Govindarajan et al. (2023) further interpret DINO through a von Mises-Fisher mixture on the unit sphere. MP also uses a class-wise mean, but in a different space and role: the mean is estimated from teacher predictive distributions after the square-root transformation, and its direction defines the axis for sample-level projection matching, thereby relaxing the supervision.

*Relations among samples.* A third line of work defines the learning objective through pairwise or triplet relations between samples, such as correlations, distances, or rankings. Relation-based distillation (Park et al., 2019; Peng et al., 2019) and metric learning objectives, including the classical triplet loss (Schroff et al., 2015) and the multi-similarity loss (Wang et al., 2019), learn such sample-to-sample relations, usually in feature space. This provides another way to move beyond exact pointwise targets, whereas MP does so through a subset summary of output distributions, without introducing pairwise or triplet relations.

## 3 Method

MP avoids exact pointwise imitation of the teacher's per-sample probability vectors, which may transfer sample-specific unreliable signals together with useful knowledge. Instead, MP distills class-wise structure by decomposing the teacher predictions into a shared subset direction and sample-level contributions to that direction. We now define this decomposition on the square-root probability sphere.

### 3.1 Setup

Let $f^t$ and $f^s$ denote the teacher and student networks. For a mini-batch $\{(x_i, y_i)\}_{i=1}^N$ in a $K$-class problem, they produce logits $z_i^t, z_i^s \in \mathbb{R}^K$. With temperature $T > 0$, define

$$p_i^t = \text{softmax}\left(\frac{z_i^t}{T}\right), \qquad p_i^s = \text{softmax}\left(\frac{z_i^s}{T}\right). \tag{1}$$

We map these distributions to the unit positive sphere by the coordinate-wise square-root transformation

$$s_i^t = \sqrt{p_i^t}, \qquad s_i^s = \sqrt{p_i^s}. \tag{2}$$

Hence $s_i^t, s_i^s \in \mathbb{S}_+^{K-1}$. On this sphere, the inner product

$$\langle s_i^t, s_i^s \rangle = \sum_{j=1}^K \sqrt{p_{i,j}^t \, p_{i,j}^s} \tag{3}$$

is the Bhattacharyya kernel (Jebara & Kondor, 2003; Jebara et al., 2004), a classical similarity between probability measures. One-hot predictions map to orthogonal coordinate axes of $\mathbb{S}_+^{K-1}$, so the class regions and decision boundaries are fixed geometrically (Section B.1).

For each mini-batch, samples are partitioned into class-wise subsets $\{S_k\}_{k=1}^K$, where $S_k = \{i : y_i = k\}$. For each nonempty subset, we compute the empirical means of these sphere points,

$$m_k^t = \frac{1}{|S_k|} \sum_{i \in S_k} s_i^t, \qquad m_k^s = \frac{1}{|S_k|} \sum_{i \in S_k} s_i^s, \tag{4}$$

and normalize them to obtain the teacher and student mean directions

$$\mu_k^t = \frac{m_k^t}{\|m_k^t\|_2}, \qquad \mu_k^s = \frac{m_k^s}{\|m_k^s\|_2}. \tag{5}$$

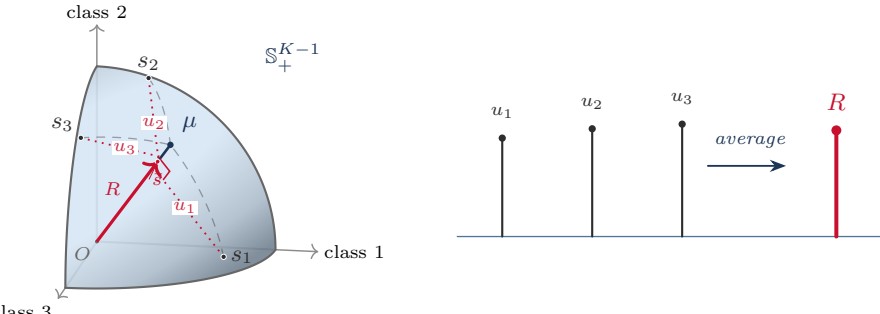

Figure 2: **The subset mean vector decomposes into a direction and a length. Left.** On the unit positive sphere $\mathbb{S}^{K-1}_+$ ($K = 3$ shown), the three samples are drawn scattered around the subset mean direction $\mu$. They define a subset mean vector $\bar{s}$, whose normalized direction is $\mu$. Each sample has a scalar projection $u_i = \mu^\top s_i$ onto this direction. The resultant length $R = \|\bar{s}\|_2$ equals the average of these projections. **Right.** The same relation in scalar form: the sample-level projections $u_i$ average to the subset-level length $R$. Mean matching aligns $\mu$, while projection matching aligns the sample-level coordinates $u_i$ and therefore also the resultant length.

## 3.2 From mean vectors to matching objectives

The teacher subset mean vector provides two quantities: its normalized direction $\mu_k^t$ and its length. The latter is the resultant length (Mardia & Jupp, 1999; Banerjee et al., 2005),

$$R_k^t = \|m_k^t\|_2. \tag{6}$$

Since $\mu_k^t = m_k^t/\|m_k^t\|_2$, this length can be written as the average projection of teacher predictions onto $\mu_k^t$:

$$R_k^t = (\mu_k^t)^\top m_k^t = \frac{1}{|S_k|} \sum_{i \in S_k} u_{k,i}^t, \tag{7}$$
$$u_{k,i}^t = (\mu_k^t)^\top s_i^t.$$

Thus the subset mean provides two supervision targets: a subset-level mean direction $\mu_k^t$ and a sample-level projection $u_{k,i}^t$ for each teacher prediction. Figure 2 visualizes this decomposition. Further geometric and statistical details are given in Sections B.2, B.3 and B.5.

### 3.2.1 Mean and projection matching

MP turns the two objects above into two matching terms. The mean-matching loss is defined as:

$$\mathcal{L}_{\text{mean}}^{(k)} = 1 - \langle \mu_k^t, \mu_k^s \rangle,$$
$$\mathcal{L}_{\text{mean}} = \frac{1}{K_B} \sum_{k:\, |S_k|>0} \mathcal{L}_{\text{mean}}^{(k)}, \tag{8}$$

where $K_B = |\{k : |S_k| > 0\}|$ is the number of nonempty classes in the mini-batch. With $u_{k,i}^s = (\mu_k^t)^\top s_i^s$, the projection-matching loss is defined as:

$$\mathcal{L}_{\text{proj}}^{(k)} = \sqrt{\frac{1}{|S_k|} \sum_{i \in S_k} \left( u_{k,i}^t - u_{k,i}^s \right)^2},$$
$$\mathcal{L}_{\text{proj}} = \frac{1}{K_B} \sum_{k:\, |S_k|>0} \mathcal{L}_{\text{proj}}^{(k)}. \tag{9}$$

Both terms are nonnegative and bounded in $[0, 1]$. This boundedness comes from the square-root transformation to the unit positive sphere and the decomposition of the subset mean vector.

The MP distillation loss is their sum,

$$\mathcal{L}_{\mathrm{MP}} = \mathcal{L}_{\mathrm{mean}} + \mathcal{L}_{\mathrm{proj}}. \tag{10}$$

### 3.2.2 Full objective

The MP distillation loss is combined with the standard cross-entropy loss,

$$\mathcal{L}_{\mathrm{CE}} = -\frac{1}{N} \sum_{i=1}^{N} \log q_{i,y_i}, \qquad q_i = \mathrm{softmax}(z_i^s), \tag{11}$$

giving the full training objective

$$\mathcal{L} = (1 - \alpha)\mathcal{L}_{\mathrm{CE}} + \alpha T \, \mathcal{L}_{\mathrm{MP}}, \qquad \alpha \in [0, 1]. \tag{12}$$

The factor $T$ is motivated by the explicit $1/T$ logit-to-square-root Jacobian; higher-order effects are discussed in Section B.8. The admissible-set interpretation in Figure 1 applies to the MP distillation term, while cross-entropy anchors the student to the gold labels.

### 3.2.3 Properties and implementation

MP has a simple single-sample endpoint. When $|S_k| = 1$, the subset mean direction coincides with the single sample, so $\mu_k^t = s_i^t$ and $\mu_k^s = s_i^s$. Both per-subset terms then reduce to the same pointwise term,

$$\mathcal{L}_{\mathrm{mean}}^{(k)} = \mathcal{L}_{\mathrm{proj}}^{(k)} = 1 - \langle s_i^t, s_i^s \rangle = H^2(p_i^t, p_i^s). \tag{13}$$

Thus each MP term is exactly the squared Hellinger distance, and the summed MP loss is the constant multiple $2H^2(p_i^t, p_i^s)$. We evaluate this single-sample endpoint on CIFAR-100 (Krizhevsky, 2009) in Table E.12.

The computation follows directly from the definition. Within each mini-batch, we group samples by gold label, form the teacher and student subset means of these sphere points, and compute $u_{k,i}^t$ and $u_{k,i}^s$ by inner products with $\mu_k^t$. Empty subsets are skipped and singleton subsets use the same formulas. The pseudocode is given in Figure D.1. Temperature enters only through $p = \mathrm{softmax}(z/T)$: we use $T = 2$ for MP and $T = 4$ for KL baselines, with sensitivity and gradient analyses in Sections B.7, B.8 and 4.4.

## 4 Experiments

We evaluate MP from four perspectives. Together, the experiments test how MP changes what the student inherits from the teacher. The evaluation is mechanism-focused and targets imperfect-teacher fine-tuning. *(i) NLI under an imperfect teacher (Section 4.1)*: a diagnostic case where an MNLI-trained teacher is applied to shifted and adversarial NLI datasets. *(ii) Fine-grained image classification (Section 4.2)*: a setting where the teacher's argmax is usually correct, but visually similar classes can make non-target probabilities noisy. *(iii) Per-sample mechanism analysis (Section 4.3)*: we measure how often the student recovers teacher mistakes, mimics them, or damages teacher-correct predictions. *(iv) Analysis and ablation (Section 4.4)*: distribution shift robustness, non-target teacher information, MP design ablations, the single-sample Hellinger endpoint, and cross-backbone consistency. Unless otherwise stated, baseline hyperparameters follow the original papers.

### 4.1 NLI distillation under an imperfect teacher

We use NLI as a diagnostic case study for imperfect-teacher distillation. A teacher trained on MNLI (Williams et al., 2018) is applied directly to shifted or adversarial NLI datasets, creating a setting where its predictions can be unreliable. We use a frozen `roberta-large-mnli` teacher (Liu et al., 2020) on several three-class NLI datasets: ANLI-R1/R2/R3 (Nie et al., 2020), FEVER-NLI (Thorne et al., 2018), and WANLI (Liu et al., 2022). Across these shifted and adversarial three-class NLI datasets, teacher accuracy ranges from 25.0% to 61.4%. We distill into two student encoders: `electra-small-discriminator` (Clark et al., 2020) and `distilbert-base-uncased` (Sanh et al., 2020). All students are trained with AdamW, batch size 32, three

Table 1: NLI distillation with an imperfect teacher. We report student top-1 accuracy↑ (%, mean ± std over three seeds). ANLI students are trained on pooled R1+R2+R3 and evaluated on each round separately. **Bold** indicates the best student result within each student backbone in each column.

| S | Method | FEVER-NLI | WANLI | ANLI-R1 | ANLI-R2 | ANLI-R3 | *ANLI avg* |
|---|---|---|---|---|---|---|---|
| | *Teacher*: roberta-large-mnli | 46.2 | 61.4 | 44.6 | 27.1 | 25.0 | *32.2* |
| ELECTRA-small | CE | **73.4** ±0.1 | **67.0** ±0.2 | **50.5** ±0.2 | **39.5** ±0.8 | **42.1** ±1.1 | ***44.0*** |
| | KD | 48.3 ±0.4 | 56.1 ±0.2 | 36.4 ±0.9 | 33.7 ±1.0 | 32.8 ±0.8 | *34.3* |
| | DKD | 51.9 ±0.3 | 56.8 ±0.4 | 34.4 ±1.1 | 31.9 ±1.1 | 31.6 ±0.7 | *32.6* |
| | RLD | 73.0 ±0.0 | 66.4 ±0.5 | 43.2 ±0.9 | 36.3 ±1.3 | 35.8 ±0.8 | *38.4* |
| | MP | 59.6 ±5.7 | 58.8 ±1.2 | 43.5 ±1.2 | 37.2 ±0.7 | 36.8 ±1.0 | *39.2* |
| DistilBERT | CE | 73.6 ±0.1 | **67.7** ±0.1 | **52.6** ±1.3 | **42.0** ±0.2 | **44.6** ±0.1 | ***46.4*** |
| | KD | 48.2 ±0.3 | 57.3 ±0.6 | 38.2 ±0.8 | 33.7 ±0.4 | 33.2 ±1.2 | *35.0* |
| | DKD | 52.3 ±0.2 | 57.8 ±0.4 | 36.2 ±0.8 | 33.1 ±0.6 | 32.9 ±0.5 | *34.1* |
| | RLD | **73.6** ±0.1 | 67.3 ±0.6 | 47.5 ±1.6 | 39.1 ±0.2 | 38.9 ±0.5 | *41.8* |
| | MP | 56.7 ±0.9 | 59.0 ±0.1 | 44.5 ±0.3 | 38.1 ±0.8 | 37.9 ±1.1 | *40.2* |

epochs, and three seeds. We compare CE, KD (Hinton et al., 2015), DKD (Zhao et al., 2022), RLD (Sun et al., 2025), and MP. KD and MP use outer $\alpha = 0.9$. For DKD and RLD, we use their published defaults. The KL baselines, KD, DKD, and RLD, use $T = 4$, while MP uses $T = 2$, following the discussion in Section 3.2.

Table 1 confirms that this is a difficult teacher setting: the MNLI teacher is weak under shift and adversarial evaluation, and KD and DKD often inherit the teacher's errors. On FEVER-NLI and WANLI, RLD is the strongest distillation method. This is consistent with its label-aware design in a three-class setting, where masking classes ranked above the gold label can strongly reshape the target. MP is weaker than RLD on these two splits but remains well above KD and DKD.

The adversarial ANLI splits show a different pattern. MP is the strongest distillation method for the weaker ELECTRA-small student on ANLI average, while RLD is stronger for the larger DistilBERT student; in both cases, MP and RLD are far above KD and DKD.

## 4.2 Fine-grained distillation

Even when the teacher's argmax prediction is usually correct, its non-target probabilities can still be noisy when classes are visually similar. In this setting, imitating these non-target probabilities pointwise, or emphasizing them, may not help the student and can come at a calibration cost. We therefore evaluate three fine-grained datasets: CUB-200 (Welinder et al., 2010), Stanford Cars (Krause et al., 2013), and Oxford-IIIT Pets (Parkhi et al., 2012). Inspired by current vision-transformer distillation practice (Habib et al., 2023), we use a pretrained-backbone fine-tuning protocol: a teacher with supervised or self-supervised pretraining is first fine-tuned on the target dataset and then distilled into a smaller pretrained student under a shared training schedule.

We use two teacher families, ViT-Base/16 (Dosovitskiy et al., 2021) and DINOv2-Base/14 (Oquab et al., 2024), and pair each with two students, DeiT-Tiny (Touvron et al., 2021) and ViT-Tiny (Dosovitskiy et al., 2021), giving four teacher-student backbones per dataset and twelve in total. Teacher checkpoints are selected by validation NLL with early stopping; student checkpoints by validation top-1. All students are trained with AdamW, batch size 256, 50 epochs, warmup plus cosine decay, and three seeds; full optimization details are given in Section D.1. This batch size keeps the expected class-wise subset size above one even for $K \approx 200$.

We compare MP with CE, KD, DKD, and RLD, using the same scaling and weighting conventions as in Section 4.1. KD and MP use outer $\alpha = 0.9$. For DKD and RLD, we use their published defaults. All KL baselines use $T = 4$, and MP uses $T = 2$. In the main paper, we report top-1 accuracy and ECE; extended metrics and MP ablations are summarized in Section 4.4, with full details in Section D.1.

Table 2: Fine-grained classification under the 50-epoch fine-tuning protocol. We evaluate three datasets (CUB-200, Stanford Cars, Oxford-IIIT Pets), two teachers (ViT-B/16, DINOv2-B/14), and two students (DeiT-Tiny, ViT-Tiny). T1 is top-1 accuracy and ECE is expected calibration error, both in %. Results are mean $\pm$ std over three seeds. **Bold** marks the best distillation method per column; CE is shown as a reference. DKD/RLD use published defaults; DKD $\beta$-sensitivity is reported in Table 5.

| | *teacher: ViT-B/16* | | | | *teacher: DINOv2-B/14* | | | |
| | DeiT-Tiny | | ViT-Tiny | | DeiT-Tiny | | ViT-Tiny | |
| Method | T1$\uparrow$ | ECE$\downarrow$ | T1$\uparrow$ | ECE$\downarrow$ | T1$\uparrow$ | ECE$\downarrow$ | T1$\uparrow$ | ECE$\downarrow$ |
|---|---|---|---|---|---|---|---|---|
| *CUB-200* | (teacher top-1: *85.65* for ViT-B/16, *88.69* for DINOv2-B/14) | | | | | | | |
| CE | $75.90_{\pm0.74}$ | $9.44_{\pm0.69}$ | $79.96_{\pm0.22}$ | $6.20_{\pm1.07}$ | $75.90_{\pm0.74}$ | $9.44_{\pm0.69}$ | $79.96_{\pm0.22}$ | $6.20_{\pm1.07}$ |
| KD | $77.79_{\pm0.35}$ | $18.02_{\pm0.68}$ | $81.75_{\pm0.02}$ | $20.67_{\pm0.26}$ | $78.45_{\pm0.41}$ | $18.94_{\pm1.05}$ | $82.06_{\pm0.32}$ | $24.32_{\pm0.47}$ |
| DKD | $78.02_{\pm0.33}$ | $25.85_{\pm0.61}$ | $81.33_{\pm0.41}$ | $28.83_{\pm0.70}$ | $\mathbf{78.87}_{\pm0.22}$ | $26.30_{\pm0.78}$ | $80.66_{\pm0.99}$ | $35.34_{\pm1.55}$ |
| RLD | $\mathbf{78.73}_{\pm0.17}$ | $19.49_{\pm0.98}$ | $\mathbf{82.78}_{\pm0.21}$ | $21.82_{\pm0.37}$ | $78.83_{\pm0.48}$ | $19.57_{\pm1.17}$ | $\mathbf{82.54}_{\pm0.32}$ | $24.44_{\pm0.41}$ |
| MP | $77.83_{\pm0.15}$ | $\mathbf{11.17}_{\pm0.47}$ | $82.44_{\pm0.18}$ | $\mathbf{14.99}_{\pm0.21}$ | $77.72_{\pm0.65}$ | $\mathbf{12.54}_{\pm0.82}$ | $82.23_{\pm0.10}$ | $\mathbf{15.74}_{\pm0.06}$ |
| *Stanford Cars* | (teacher top-1: *82.56* for ViT-B/16, *92.37* for DINOv2-B/14) | | | | | | | |
| CE | $72.50_{\pm0.11}$ | $13.54_{\pm0.36}$ | $68.53_{\pm0.67}$ | $7.87_{\pm0.12}$ | $72.50_{\pm0.11}$ | $13.54_{\pm0.36}$ | $68.53_{\pm0.67}$ | $7.87_{\pm0.12}$ |
| KD | $75.03_{\pm0.29}$ | $25.93_{\pm0.68}$ | $71.88_{\pm0.58}$ | $30.01_{\pm0.32}$ | $74.42_{\pm0.38}$ | $24.66_{\pm0.51}$ | $69.97_{\pm0.92}$ | $32.53_{\pm0.35}$ |
| DKD | $74.30_{\pm0.13}$ | $35.43_{\pm0.86}$ | $69.20_{\pm0.44}$ | $40.40_{\pm0.20}$ | $74.08_{\pm0.54}$ | $34.09_{\pm0.65}$ | $64.21_{\pm0.86}$ | $42.53_{\pm0.25}$ |
| RLD | $\mathbf{75.89}_{\pm0.37}$ | $27.40_{\pm0.62}$ | $72.67_{\pm0.47}$ | $31.67_{\pm0.31}$ | $\mathbf{75.43}_{\pm0.59}$ | $26.21_{\pm0.66}$ | $71.23_{\pm0.68}$ | $34.24_{\pm0.16}$ |
| MP | $75.62_{\pm0.75}$ | $\mathbf{15.55}_{\pm0.66}$ | $\mathbf{73.61}_{\pm0.46}$ | $\mathbf{18.10}_{\pm0.16}$ | $75.37_{\pm0.38}$ | $\mathbf{15.76}_{\pm0.38}$ | $\mathbf{73.19}_{\pm0.64}$ | $\mathbf{19.30}_{\pm0.46}$ |
| *Oxford-IIIT Pets* | (teacher top-1: *93.43* for ViT-B/16, *95.12* for DINOv2-B/14) | | | | | | | |
| CE | $91.22_{\pm0.23}$ | $1.63_{\pm0.24}$ | $89.59_{\pm0.15}$ | $2.15_{\pm1.65}$ | $91.22_{\pm0.23}$ | $1.63_{\pm0.24}$ | $89.59_{\pm0.15}$ | $2.15_{\pm1.65}$ |
| KD | $91.54_{\pm0.30}$ | $8.45_{\pm0.38}$ | $90.91_{\pm0.29}$ | $12.58_{\pm0.26}$ | $91.78_{\pm0.30}$ | $7.25_{\pm0.51}$ | $90.70_{\pm0.17}$ | $11.31_{\pm0.53}$ |
| DKD | $91.56_{\pm0.04}$ | $9.63_{\pm0.30}$ | $90.88_{\pm0.34}$ | $15.02_{\pm0.08}$ | $91.83_{\pm0.32}$ | $7.07_{\pm1.34}$ | $90.89_{\pm0.34}$ | $13.95_{\pm0.18}$ |
| RLD | $\mathbf{91.64}_{\pm0.11}$ | $\mathbf{6.19}_{\pm0.68}$ | $\mathbf{91.11}_{\pm0.25}$ | $\mathbf{10.99}_{\pm0.23}$ | $\mathbf{92.03}_{\pm0.40}$ | $\mathbf{5.46}_{\pm1.10}$ | $\mathbf{91.01}_{\pm0.24}$ | $\mathbf{10.53}_{\pm0.65}$ |
| MP | $91.47_{\pm0.28}$ | $6.92_{\pm0.43}$ | $90.71_{\pm0.27}$ | $11.32_{\pm0.26}$ | $91.62_{\pm0.34}$ | $5.84_{\pm0.42}$ | $90.62_{\pm0.15}$ | $10.61_{\pm0.11}$ |

Table 2 shows that MP has a clear calibration advantage among distillation methods. Averaged over all twelve settings, MP attains the lowest ECE among distillation methods (13.2%, versus 19.6, 26.2, and 19.8 for KD, DKD, and RLD) while its top-1 is on par with the strongest baseline (81.9 versus RLD's 82.0); see Table E.5 for the full mean summary. The gains are strongest on the harder fine-grained datasets, where high inter-class similarity can make non-target probabilities less reliable. On Stanford Cars, MP gives the best ECE among distillation methods on all four backbones, and matches the strongest baseline (RLD) on top-1. On CUB-200, MP gives the best ECE among distillation methods on all four backbones and remains competitive on top-1. By contrast, on Oxford-IIIT Pets, the differences are small: all distillation methods are nearly tied in top-1 and only slightly above the CE baseline. Overall, distillation improves top-1 over CE in most cases, but often at a calibration cost; MP reduces this cost while preserving the accuracy benefit.

DKD has large calibration errors on CUB-200 and Stanford Cars, suggesting that emphasizing the non-target distribution can be less reliable in this fine-grained setting; the effect is smaller on Pets. The DKD $\beta$-sweep in Table 5 further supports this interpretation: smaller $\beta_{\text{dkd}}$ consistently lowers ECE.

### 4.3 Per-sample mechanism analysis

Beyond top-1 and calibration, we analyze what each student inherits from the teacher at the sample level. To compare supervision objects, we use KD as the direct pointwise KL counterpart to MP. On the test set, conditioned on the gold label $y_i = k$, we report the *recovery rate* $\rho = \mathbb{P}(\text{student} = k \mid \text{teacher} \neq k)$, the *damage rate* $\delta = \mathbb{P}(\text{student} \neq k \mid \text{teacher} = k)$, and the *mimicry rate* $\omega = \mathbb{P}(\text{student} = \text{teacher} \mid \text{teacher} \neq k)$. Recovery measures how often the student corrects a teacher mistake, damage measures how often it breaks a teacher-correct prediction, and mimicry measures how often it copies the teacher's wrong answer. The

Table 3: Per-sample mechanism analysis on fine-grained classification. The first three rows show representative teacher-student pairs, one per dataset; the Mean row averages over all twelve fine-grained backbones. Full per-backbone results and the damage rate $\delta$ are in Section E.3. $\rho$ and $\omega$ are recovery and mimicry; ECE is reproduced from Table 2. For the representative rows, CUB-200 and Stanford Cars use ViT-B/16 $\rightarrow$ DeiT-Tiny, and Oxford Pets uses DINOv2-B/14 $\rightarrow$ DeiT-Tiny. Student values are mean $\pm$ std over three seeds.

| Dataset | ECE (%)$\downarrow$ | | $\rho$ (%)$\uparrow$ | | $\omega$ (%)$\downarrow$ | |
| | KD | MP | KD | MP | KD | MP |
|---|---|---|---|---|---|---|
| CUB-200 | 18.02 $\pm$0.68 | **11.17** $\pm$0.47 | 26.70 $\pm$0.78 | **30.88** $\pm$0.84 | 39.55 $\pm$0.30 | **35.42** $\pm$0.78 |
| Stanford Cars | 25.93 $\pm$0.68 | **15.55** $\pm$0.66 | 31.34 $\pm$1.93 | **33.70** $\pm$1.68 | 28.46 $\pm$1.51 | **24.70** $\pm$0.73 |
| Oxford Pets | 7.25 $\pm$0.51 | **5.84** $\pm$0.42 | 27.56 $\pm$1.71 | **32.59** $\pm$1.29 | 59.40 $\pm$1.41 | **51.40** $\pm$2.56 |
| Mean | 19.56 | **13.15** | 27.41 | **30.35** | 44.62 | **40.50** |

Table 4: Office-Home OOD$-$ID NLL gap (domain-averaged over four source domains and three seeds; lower is better) for DeiT-Tiny and ViT-Tiny. Numbers are extracted from the last column of Table E.1; **bold** marks the best distillation method per student. Full results, including DINOv2-S/14, are reported in Table E.1.

| Student | CE | KD | DKD | RLD | MP |
|---|---|---|---|---|---|
| DeiT-Tiny | 1.492 | 1.202 | 1.238 | 1.239 | **1.160** |
| ViT-Tiny | 1.762 | 1.269 | 1.371 | 1.313 | **1.213** |

mimicry metric is inspired by the teacher-student agreement analysis of Stanton et al. (2021). KD and MP are evaluated on the same teacher-wrong partition for $\rho, \omega$ and the same teacher-right partition for $\delta$.

Table 3 shows a consistent pattern on all three highlighted backbones. MP recovers more teacher-wrong samples than KD (+4.2 pp on CUB-200, +2.4 pp on Cars, and +5.0 pp on Pets), while damage $\delta$ remains nearly unchanged (within 1 pp on all three, Section E.3). MP also has a lower mimicry rate, by 4.1, 3.8, and 8.0 pp, respectively. At comparable top-1, this is consistent with MP relaxing pointwise copying: the student is left less tied to the teacher's exact wrong prediction and more able to recover the gold label. The mean row of Table 3 shows the same tendency across all backbones, with $\rho$ higher by 2.9 pp and $\omega$ lower by 4.1 pp; damage $\delta$ is essentially unchanged ($-0.3$ pp). Per-backbone results are reported in Section E.3.

## 4.4 Analysis and ablation

We use this section for additional analysis beyond the main results: distribution shift robustness, a baseline-side probe of noisy non-target information in fine-grained distillation, MP design ablations, the single-sample Hellinger endpoint, and further statistical and mechanism analyses of the experimental results.

**Robustness under distribution shift.** We evaluate on Office-Home (Venkateswara et al., 2017), which contains four visually distinct domains and $K = 65$ classes. For each source domain, we fine-tune a ViT-B/16 teacher and distill three students (DeiT-Tiny, ViT-Tiny, DINOv2-S/14) on that source, evaluating on the union of the other three domains as OOD. Because top-1 differences between distillation methods are small on this benchmark, we summarize robustness through the domain-average OOD$-$ID NLL gap, a single shift-robustness scalar (Table 4). All distillation methods substantially reduce the OOD$-$ID NLL gap relative to CE, and MP gives the smallest gap for DeiT-Tiny (1.160) and ViT-Tiny (1.213), although other Office-Home metrics, including calibration metrics, are mixed. Full results for DINOv2-S/14, OOD ECE, OOD Brier, worst-source variants, and per-source breakdowns are reported in Section E.1.

**Probing non-target information with the DKD coefficient.** DKD explicitly separates the target and non-target terms, so its non-target coefficient $\beta_{dkd}$ provides a useful probe of teacher non-target information in the fine-grained setting. The $\beta$-sweep shows a clear pattern on the fine-grained grid (Table 5): smaller $\beta_{dkd}$ consistently improves calibration, and in the hardest Cars settings the published default can also hurt top-1. This suggests that emphasizing the non-target distribution can be less reliable on visually similar

Table 5: DKD $\beta_{\mathrm{dkd}}$ sweep on the two harder fine-grained datasets, CUB-200 and Stanford Cars. The default $\beta_{\mathrm{dkd}} = 8$ corresponds to the values in Table 2; $\beta_{\mathrm{dkd}} \in \{2, 4\}$ are additional sweep points. Results are averaged over three seeds. The Oxford-IIIT Pets results are reported in Table E.13.

| Dataset | Backbone | $\beta_{\mathrm{dkd}} = 2$ | | $\beta_{\mathrm{dkd}} = 4$ | | $\beta_{\mathrm{dkd}} = 8$ (default) | |
|---|---|---|---|---|---|---|---|
| | | T1↑ | ECE↓ | T1↑ | ECE↓ | T1↑ | ECE↓ |
| CUB-200 | ViT-B/16 → DeiT-Tiny | 78.41 | **15.59** | 78.71 | 19.23 | 78.02 | 25.85 |
| | ViT-B/16 → ViT-Tiny | 82.91 | **16.99** | 82.73 | 21.52 | 81.33 | 28.83 |
| | DINOv2-B/14 → DeiT-Tiny | 78.13 | **16.35** | 78.81 | 19.32 | 78.87 | 26.30 |
| | DINOv2-B/14 → ViT-Tiny | 82.58 | **18.28** | 82.53 | 24.73 | 80.66 | 35.34 |
| Stanford Cars | ViT-B/16 → DeiT-Tiny | 75.85 | **21.86** | 75.85 | 27.40 | 74.30 | 35.43 |
| | ViT-B/16 → ViT-Tiny | 73.17 | **24.12** | 72.67 | 31.66 | 69.20 | 40.40 |
| | DINOv2-B/14 → DeiT-Tiny | 75.24 | **21.33** | 75.46 | 26.26 | 74.08 | 34.09 |
| | DINOv2-B/14 → ViT-Tiny | 72.65 | **25.99** | 71.22 | 34.25 | 64.21 | 42.53 |

fine-grained classes. The result is consistent with MP's motivation of avoiding pointwise imitation of the full teacher distribution.

**MP design ablations.**  We run four MP ablations on two fine-grained backbones (CUB-200 and Stanford Cars, both ViT-B/16 → ViT-Tiny). *(i) Batch size* (Table E.9): we compare fixed-epochs and fixed-updates protocols to separate the effect of class-wise subset size from changes in the number of optimizer updates. *(ii) MP temperature* (Table E.8): the sweep over $T \in \{1, 2, 4\}$ shows stronger calibration at $T = 1$ but better top-1 at $T = 2$. *(iii) Subset choice* (Table E.10): class-wise grouping performs similarly to teacher-argmax grouping and outperforms batch-level summary variants. *(iv) Loss components* (Table E.11): the full objective gives the best top-1, whereas using only the mean loss or only the projection loss can improve ECE but does not achieve the same overall balance.

**Cross-backbone consistency.**  Across all twelve fine-grained settings, MP shows a stable advantage on the metrics targeted by the proposed supervision object. For calibration, MP reduces ECE relative to KD by 6.40 pp ($[−9.19, −3.61]$, 12/12 settings), relative to DKD by 13.05 pp ($[−18.26, −7.84]$, 12/12), and relative to RLD by 6.68 pp ($[−10.33, −3.03]$, 8/12). The per-sample mechanism metrics show the same pattern: MP increases recovery $\rho$ over KD ($[+1.93, +3.95]$, 11/12) and reduces mimicry $\omega$ ($[−5.44, −2.81]$, 12/12), while the damage gap $\delta$ remains centered near zero ($[−1.02, +0.50]$, 6/12; Section E.3). At the same time, MP keeps top-1 accuracy comparable to KD, with a small positive mean difference ($+0.51$ pp, $[−0.17, +1.19]$, 7/12). Thus, across backbones, MP improves the quality of teacher inheritance: it keeps accuracy comparable while improving calibration, increasing recovery, and reducing mimicry.

**Single-sample Hellinger endpoint.**  When each nonempty subset satisfies $|S_k| = 1$, each MP term reduces to the squared Hellinger distance and the summed loss differs only by a constant factor. We therefore evaluate this single-sample objective on CIFAR-100 from scratch under the standard CRD/`mdistiller` protocol (Tian et al., 2020). In this setting, it remains comparable to KD (Table E.12).

**Mechanism decomposition: geometry and subset relaxation.**  We further decompose the mechanism results by using the single-sample endpoint as a pivot. This decomposition uses bs $= 512$ with the 50-epoch schedule and a learning rate scaled with batch size (Goyal et al., 2017). The contrast [single-sample − KD] measures the effect of replacing pointwise KL matching by pointwise square-root geometry. The contrast [MP − single-sample] then measures the effect added by the subset relaxation under the same square-root geometry.

Table 6(a) shows that the two contrasts emphasize different parts of the effect. The single-sample endpoint already accounts for most of MP's ECE reduction over KD, suggesting that much of the calibration gain comes from the square-root geometry of the MP construction. It also improves recovery and mimicry, so these mechanism changes are not caused by subset relaxation alone. MP further changes these inheritance

Table 6: Mechanism decomposition using the single-sample endpoint as a pivot. **(a)** Paired differences over the twelve fine-grained settings at $bs = 512$ (mean $\pm$ 95% CI); **bold** marks intervals excluding zero. The two contrasts sum to the total $[MP - KD]$ difference. **(b)** Subset gap $[MP - single\text{-}sample]$ by dataset and batch size, with expected subset size $\mathbb{E}[|S|] = bs/K$. The $bs = 256$ entries are shown only as aggregate point estimates; for $bs = 512$, bold entries indicate 95% CIs that exclude zero.

*(a) Pointwise geometry and subset decomposition (bs = 512, 12 settings)*

| Contrast | $\Delta\text{ECE}\downarrow$ | $\Delta\rho\uparrow$ | $\Delta\omega\downarrow$ | $\Delta\text{top-1}\uparrow$ |
|---|---|---|---|---|
| Pointwise square-root [single-sample − KD] | $-\mathbf{7.38}_{\pm3.20}$ | $+\mathbf{2.32}_{\pm0.71}$ | $-\mathbf{2.27}_{\pm0.59}$ | $+\mathbf{1.51}_{\pm1.44}$ |
| Subset [MP − single-sample] | $-\mathbf{0.75}_{\pm0.47}$ | $+\mathbf{0.97}_{\pm0.68}$ | $-\mathbf{1.71}_{\pm1.04}$ | $-\mathbf{0.35}_{\pm0.09}$ |
| Total [MP − KD] | $-\mathbf{8.13}_{\pm3.51}$ | $+\mathbf{3.28}_{\pm0.91}$ | $-\mathbf{3.98}_{\pm1.21}$ | $+\mathbf{1.16}_{\pm1.37}$ |

*(b) Subset gap [MP − single-sample] versus expected subset size*

| Dataset | $K$ | $\mathbb{E}[|S|]_{256/512}$ | $\Delta\rho\uparrow$ | | $\Delta\omega\downarrow$ | |
|---|---|---|---|---|---|---|
| | | | bs256 | bs512 | bs256 | bs512 |
| Oxford Pets | 37 | 6.9 / 13.8 | $+2.23$ | $+\mathbf{2.32}$ | $-3.80$ | $-\mathbf{3.65}$ |
| CUB-200 | 200 | 1.28 / 2.56 | $+0.11$ | $+\mathbf{0.51}$ | $-0.36$ | $-\mathbf{0.94}$ |
| Stanford Cars | 196 | 1.31 / 2.61 | $+0.10$ | $+0.07$ | $-0.62$ | $-0.52$ |
| Pearson $r$ with $\mathbb{E}[|S|]$ | | | $+0.90$ | | $-0.88$ | |

metrics over this endpoint: recovery $\rho$ increases and mimicry $\omega$ decreases, while damage $\delta$ remains essentially unchanged.

Table 6(b) checks whether the subset component depends on the expected class-wise subset size. The $[MP - single\text{-}sample]$ gap is large for Oxford Pets, where $K = 37$ gives much larger expected subsets. It becomes clearer on CUB-200 at batch size 512, where $\mathbb{E}[|S|] \approx 2.56$, and remains small on Stanford Cars. Across the six dataset–batch settings, the inheritance gaps follow the expected direction with $\mathbb{E}[|S|]$: recovery is positively correlated ($r = +0.90$) and mimicry is negatively correlated ($r = -0.88$). ECE reduction does not show the same subset-size dependence. Overall, this decomposition supports the mechanism interpretation: the pointwise endpoint already accounts for much of the ECE reduction and a substantial part of the observed changes in recovery and mimicry, while the subset relaxation adds a size-dependent effect most visible in teacher-error inheritance; both arise from the same square-root probability construction.

## 5 Conclusion and discussion

We proposed MP, a logit-based distillation objective that changes the object of supervision. Instead of matching the teacher's full predictive distribution sample by sample, MP maps the predictions to the square-root probability sphere and uses the empirical mean vector of each class-wise subset: its direction is matched at the subset level, and its length is transferred sample by sample through scalar projections along that direction. The resulting objective has two bounded terms, mean matching and projection matching; in the single-sample limit, each term equals the squared Hellinger distance. MP is therefore not a different distance to the same teacher point. It requires the student to preserve the subset-level direction and one teacher-specific projection coordinate per sample, while cross-entropy can be viewed as further selecting among the remaining degrees of freedom. This gives an admissible-set view of distillation, in which teacher information constrains part of the student's prediction without requiring exact pointwise imitation.

Across the evaluated settings, MP is most relevant when teacher predictions are informative but not fully reliable. The NLI diagnostic illustrates this imperfect-teacher setting, while the fine-grained results show the clearest gains on visually similar datasets. The per-sample mechanism analysis is consistent with the testable prediction: MP recovers more teacher-wrong samples and copies teacher mistakes less often than KD, with little change in the damage rate.

One important point is that the Hellinger endpoint is not a trivial calibration add-on. It arises after the MP construction: gold-label subset partition, square-root sphere mapping, the empirical subset mean vector, and the decomposition into direction and sample-specific projection. The bounded terms and the single-sample Hellinger limit are therefore consequences of the supervision object we choose. In our experiments, the same construction shows two complementary effects: the single-sample endpoint already captures much of the calibration benefit and a large part of the recovery and mimicry change, while MP's subset relaxation adds a further teacher-error inheritance effect. Extended discussion and limitations are provided in Section A.

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

## Appendix outline

The appendix provides additional discussion, derivations, and experimental details. It is organized as follows.

- Section A: extended discussion of design choices, limiting cases, empirical scope, and limitations.

- Section B: theoretical analysis of the square-root geometry, the empirical class-wise subset statistics, the two MP losses, a von Mises-Fisher parametric lens, the single-sample Hellinger endpoint, temperature effects, and gradient scaling.

- Section C: the admissible-set geometry induced by the projection constraint on the unit positive sphere, and its relation to class regions.

- Section D: experimental settings, MP implementation, hyperparameters, and baseline loss definitions.

- Section E: extended experiments and analyses, covering the full Office-Home results, the per-sample recovery analysis on the 12-backbone fine-grained grid, the MP ablations, and the single-sample endpoint.

# A  Extended discussion

## A.1  Design choices

The square-root transformation maps predictive distributions to the unit positive sphere $\mathbb{S}_+^{K-1}$. This is a geometric representation that follows from the simplex constraint. It also induces class regions through the argmax partition. The directional-statistical view in MP characterizes this spherical supervision object. For a class-wise subset $S_k$, the teacher predictions have an empirical mean direction and resultant length, which are standard summaries for directional data. MP does not fit a vMF model or estimate a concentration parameter; it uses these empirical quantities directly: mean matching transfers the subset mean direction, and projection matching transfers the sample-level coordinate along that direction. The vMF reading of the same statistics is deferred to Section B.5.

## A.2  Scope of the theoretical analysis

The theoretical analysis in this paper characterizes the MP objective. The square-root geometry, admissible-set view, vMF-inspired lens, and single-sample Hellinger endpoint describe what MP preserves and what it relaxes: it keeps the subset-level mean direction and one sample-level projection, while leaving the remaining components less constrained. The empirical motivation is that, in some imperfect-teacher settings, the class-wise teacher summary can be more stable than individual teacher probabilities. If the class-wise mean direction is also unreliable, or if the full teacher distribution is reliable and the student can exploit it effectively, pointwise matching may be preferable.

## A.3  Limiting cases

*Single-sample subset ($|S_k| = 1$).* When each nonempty subset contains one sample, MP exactly reduces to a pointwise squared-Hellinger objective. This is the pointwise endpoint of the method.

*Low temperature ($T \downarrow 0$).* As $T$ approaches zero, individual predictions become one-hot at their argmax, and the MP subset mean direction becomes a normalized teacher-argmax histogram within $S_k$. Thus MP matches subset-level histogram structure in this limit, unlike the usual pointwise KL-based KD limit (Proposition B.2).

*Large-$K$ regime.* When $K$ is large relative to the batch size, class-wise subsets become sparse and MP approaches its single-sample endpoint. Our experiments focus on classification settings where mini-batch subsets remain meaningful; systematic large-$K$ settings are left to future work.

## A.4  Empirical scope and limitations

The empirical study focuses on the mechanism of MP across complementary settings. The NLI setting is a diagnostic case where an MNLI-trained teacher is applied to shifted and adversarial datasets. The fine-grained vision setting tests a case where visually similar classes can make non-target probabilities unreliable. Office-Home adds a domain-shift evaluation, and the single-sample CIFAR-100 experiment verifies the pointwise Hellinger endpoint.

The main experiments use pretrained-backbone fine-tuning, where both teacher and student start from strong pretrained encoders. This matches the modern vision-backbone setting studied here: distillation is applied after pretraining, and the teacher is informative but not always reliable. Traditional from-scratch protocols on CIFAR100 and ImageNet address a different setting and use batch sizes that are small relative to the number of classes, making class-wise subsets sparse. We therefore use CIFAR-100 only as a controlled check of the single-sample Hellinger endpoint.

MP is motivated by the case where the class-wise mean direction is more stable than individual teacher predictions. When this mean direction is also unstable or biased, the summary itself may be unreliable; this setting is not systematically studied here.

# B  Theoretical analysis

This appendix collects the main geometric ideas behind MP. The construction is nonparametric: predictive distributions are mapped to the unit positive sphere by the square-root transformation, class-wise subsets are summarized by empirical spherical statistics, and the loss matches a subset mean direction together with sample-level projections. The von Mises–Fisher (vMF) model is used only as an interpretive lens for these statistics, not as an assumption of the method; MP does not estimate a vMF concentration parameter.

We first recall the square-root geometry of predictive distributions (Section B.1) and define the subset mean direction, projection coordinates, and resultant length (Section B.2). We then describe the induced constraint set and derive the two MP losses (Sections B.3 and B.4). Finally, we give the vMF interpretation, prove the single-sample Hellinger limit, and discuss temperature, gradient scaling, and supporting directional-statistics identities (Sections B.5 to B.9).

## B.1  Object of interest: square-root geometry of predictive distributions

For a predictive distribution $p \in \Delta^{K-1}$, define the coordinatewise square-root transformation

$$s = \sqrt{p}. \tag{B.1}$$

Then $s_c \geq 0$ and $\sum_c s_c^2 = 1$, so $s$ lies on the unit positive sphere,

$$\mathbb{S}_+^{K-1} = \mathbb{S}^{K-1} \cap \mathbb{R}_{\geq 0}^K. \tag{B.2}$$

This is the standard square-root representation of the probability simplex. Under the convention used in this paper,

$$H^2(p, q) = 1 - \langle \sqrt{p}, \sqrt{q} \rangle \tag{B.3}$$

is the squared Hellinger distance between $p$ and $q$. Thus, probability vectors can be studied as points on the unit positive sphere without introducing any additional probabilistic model.

Throughout this appendix,

$$s_i^t = \sqrt{p_i^t}, \qquad s_i^s = \sqrt{p_i^s} \tag{B.4}$$

denote the teacher and student predictive distributions after this transformation.

## B.2  Empirical spherical statistics of a class-wise subset

For a mini-batch, let

$$S_k = \{i : y_i = k\} \tag{B.5}$$

be the subset of samples with label $k$. For a nonempty subset $S_k$, the teacher predictions define the empirical mean vector

$$m_k^t = \frac{1}{|S_k|} \sum_{i \in S_k} s_i^t. \tag{B.6}$$

Since all $s_i^t \in \mathbb{S}_+^{K-1}$, the mean is nonzero whenever $S_k$ is nonempty, and has norm $\|m_k^t\|_2 \leq 1$, with equality only when the subset predictions coincide; thus $m_k^t$ lies inside the sphere, and its length carries the subset's concentration. We therefore define the mean direction and resultant length by

$$\mu_k^t = \frac{m_k^t}{\|m_k^t\|_2}, \qquad R_k^t = \|m_k^t\|_2. \tag{B.7}$$

For each sample $i \in S_k$, define its scalar projection onto the teacher mean direction:

$$u_{k,i}^t = (\mu_k^t)^\top s_i^t. \tag{B.8}$$

The same subset mean and mean direction are defined for the student:

$$m_k^s = \frac{1}{|S_k|} \sum_{i \in S_k} s_i^s, \qquad \mu_k^s = \frac{m_k^s}{\|m_k^s\|_2}. \tag{B.9}$$

The student projection used by MP is taken along the teacher direction:

$$u_{k,i}^s = (\mu_k^t)^\top s_i^s. \tag{B.10}$$

These quantities are linked by a simple identity. Since $m_k^t = R_k^t \mu_k^t$,

$$R_k^t = (\mu_k^t)^\top m_k^t = \frac{1}{|S_k|} \sum_{i \in S_k} (\mu_k^t)^\top s_i^t = \frac{1}{|S_k|} \sum_{i \in S_k} u_{k,i}^t. \tag{B.11}$$

Thus, the average projection recovers the resultant length of the subset mean.

## B.3 From empirical statistics to a supervision object

MP uses an object derived from the teacher's sample-level predictions $s_i^t$. It preserves the subset-level teacher mean direction $\mu_k^t$ and the teacher's scalar projection $u_{k,i}^t$, while leaving directions orthogonal to $\mu_k^t$ less constrained.

For each $i \in S_k$, define

$$\mathcal{M}_{k,i}^t = \{\, s \in \mathbb{S}_+^{K-1} : (\mu_k^t)^\top s = u_{k,i}^t \,\}. \tag{B.12}$$

This set contains all positive-sphere predictions whose projection onto the teacher mean direction equals that of the teacher sample. It can be written as the intersection between the positive sphere and the affine constraint $(\mu_k^t)^\top s = u_{k,i}^t$. The teacher prediction itself belongs to this set.

In the finite-logit interior case, if $K \geq 3$ and $u_{k,i}^t < 1$, the full-sphere hyperplane slice has dimension $K-2$, and its intersection with the positive orthant contains a nontrivial neighborhood of $s_i^t$ inside that slice.

Thus the admissible set is not a single point. Its remaining degrees of freedom lie orthogonally to the teacher mean direction, so the mean-aligned component is fixed while other components remain free. The slice may also cross an argmax boundary, so that some admissible predictions peak on a class different from the teacher's top prediction; this interaction with class regions is summarized in Lemma C.1.

The MP objective enforces this construction by penalizing deviations from the following target conditions:

$$\begin{aligned} \mu_k^s &= \mu_k^t, \\ (\mu_k^t)^\top s_i^s &= u_{k,i}^t. \end{aligned} \tag{B.13}$$

The first condition aligns the subset-level mean direction, while the second matches each sample's coordinate along that direction.

## B.4 The two losses

**Mean matching.** The mean loss aligns teacher and student subset directions:

$$\mathcal{L}_{\text{mean}}^{(k)} = 1 - (\mu_k^t)^\top \mu_k^s. \tag{B.14}$$

It is nonnegative and vanishes exactly when $\mu_k^s = \mu_k^t$. Averaging over the nonempty classes in a mini-batch gives

$$\mathcal{L}_{\text{mean}} = \frac{1}{|\mathcal{K}_B|} \sum_{k \in \mathcal{K}_B} \mathcal{L}_{\text{mean}}^{(k)}, \qquad \mathcal{K}_B = \{k : |S_k| > 0\}. \tag{B.15}$$

**Projection matching.** The projection loss matches the teacher and student coordinates along $\mu_k^t$:

$$u_{k,i}^t = (\mu_k^t)^\top s_i^t, \qquad u_{k,i}^s = (\mu_k^t)^\top s_i^s. \tag{B.16}$$

For each nonempty subset,

$$\mathcal{L}_{\text{proj}}^{(k)} = \sqrt{\frac{1}{|S_k|} \sum_{i \in S_k} \left(u_{k,i}^t - u_{k,i}^s\right)^2}. \tag{B.17}$$

The batch-level projection loss is

$$\mathcal{L}_{\text{proj}} = \frac{1}{|\mathcal{K}_B|} \sum_{k \in \mathcal{K}_B} \mathcal{L}_{\text{proj}}^{(k)}. \tag{B.18}$$

This term vanishes exactly when every student prediction has the same projection as its teacher counterpart along $\mu_k^t$.

The MP loss is therefore

$$\mathcal{L}_{\text{MP}} = \mathcal{L}_{\text{mean}} + \mathcal{L}_{\text{proj}}. \tag{B.19}$$

**Proposition B.1** (Zero-loss geometry of MP)**.** *Fix a nonempty subset $S_k$. The two per-subset MP losses vanish, $\mathcal{L}_{\text{mean}}^{(k)} = \mathcal{L}_{\text{proj}}^{(k)} = 0$, if and only if*

$$\mu_k^s = \mu_k^t \qquad \text{and} \qquad (\mu_k^t)^\top s_i^s = (\mu_k^t)^\top s_i^t \quad \text{for every } i \in S_k. \tag{B.20}$$

*In this case every student prediction lies in the corresponding teacher-defined admissible set, $s_i^s \in \mathcal{M}_{k,i}^t$, and the teacher and student subsets have the same resultant length: $R_k^s = R_k^t$.*

*Proof.* The mean loss is $1 - (\mu_k^t)^\top \mu_k^s$, and both directions are unit vectors in the positive sphere. It vanishes exactly when $\mu_k^s = \mu_k^t$. The rooted projection loss is a norm of the vector of residuals $u_{k,i}^t - u_{k,i}^s$, so it vanishes exactly when every projection equality in Equation (B.20) holds. These equalities are precisely the defining constraints of $\mathcal{M}_{k,i}^t$.

Averaging the projection equalities gives

$$(\mu_k^t)^\top m_k^s = (\mu_k^t)^\top m_k^t = R_k^t. \tag{B.21}$$

Since $m_k^s = R_k^s \mu_k^s$ and $\mu_k^s = \mu_k^t$, the left-hand side is $R_k^s$. Hence $R_k^s = R_k^t$. $\qquad\square$

## B.5 vMF as one parametric lens

The vMF distribution provides one standard interpretation of the same spherical statistics. It explains why the mean direction and resultant length form a natural pair on the sphere. This subsection is not used to define the algorithm.

**Resultant length as concentration.** For a vMF model on $\mathbb{S}^{K-1}$ with parameters $(\mu, \kappa)$, the maximum-likelihood estimates from observations $\{s_i\}_{i=1}^n$ satisfy

$$\hat{\mu} = \frac{m}{\|m\|_2}, \qquad A_K(\hat{\kappa}) = R, \qquad m = \frac{1}{n} \sum_i s_i, \quad R = \|m\|_2, \tag{B.22}$$

where

$$A_K(\kappa) = \frac{I_{K/2}(\kappa)}{I_{K/2-1}(\kappa)}. \tag{B.23}$$

The function $A_K$ is continuous and strictly increasing from $[0, \infty)$ to $[0, 1)$. Hence, for $R \in [0, 1)$, the resultant length $R$ determines the MLE concentration $\hat{\kappa}$. The case $R = 1$ corresponds to the limiting regime $\hat{\kappa} = \infty$.

This relation motivates using $R$ rather than $\kappa$ as the empirical concentration statistic. Optimizing through $\kappa$ would require inverting $A_K$ and differentiating through Bessel-function quantities. MP instead works directly with projections, whose average is $R$.

**KL decomposition.** Let $f_t = \text{vMF}(\mu_t, \kappa_t)$ and $f_s = \text{vMF}(\mu_s, \kappa_s)$. Their KL divergence can be written as

$$D_{\text{KL}}(f_t \| f_s) = A_K(\kappa_t)\kappa_s\big(1 - \mu_t^\top \mu_s\big) + B_\psi(\kappa_s, \kappa_t), \tag{B.24}$$

where $\psi(\kappa) = -\log C_K(\kappa)$ is the vMF log-partition and

$$B_\psi(\kappa_s, \kappa_t) = \psi(\kappa_s) - \psi(\kappa_t) - A_K(\kappa_t)(\kappa_s - \kappa_t) \tag{B.25}$$

is the corresponding one-dimensional Bregman divergence.

This decomposition separates a directional mismatch from a concentration mismatch. In the empirical MP construction, mean matching aligns the directions $\mu_k^s$ and $\mu_k^t$. As Proposition B.1 formalizes, projection matching implies, in aggregate and under aligned directions, that

$$R_k^s = R_k^t. \tag{B.26}$$

Through the vMF MLE relation, this corresponds to matching the concentration parameter under the vMF lens. Thus, the two MP losses mirror the two empirical components that appear in the vMF interpretation: direction and resultant length.

More explicitly, if both MP terms vanish on a subset and the empirical means are nonzero, then $\mu_s = \mu_t$ and $(\mu_t)^\top s_i^s = (\mu_t)^\top s_i^t$ for every sample. Averaging the projection equalities gives $(\mu_t)^\top m_s = (\mu_t)^\top m_t$. Since $m_s = R_s\mu_s$ and $m_t = R_t\mu_t$, direction alignment implies $R_s = R_t$. The vMF MLE relation $A_K(\hat\kappa) = R$ therefore assigns the same concentration parameter to the two empirical subsets.

**Support gap.** The vMF model is defined on the full sphere, while square-root probabilities lie on the unit positive sphere. One could instead restrict the vMF density to the unit positive sphere, but its normalizing constant generally depends on both $\mu$ and $\kappa$, and the simple closed-form MLE relation is lost. We therefore use the full-sphere vMF only as a readable parametric analogy.

### B.6 Single-sample case as an exact endpoint

When $S_k = \{i\}$, the subset mean is the sample itself:

$$m_k^t = s_i^t, \qquad \mu_k^t = s_i^t, \qquad m_k^s = s_i^s, \qquad \mu_k^s = s_i^s. \tag{B.27}$$

The mean loss becomes

$$\mathcal{L}_{\text{mean}}^{(k)} = 1 - \langle s_i^t, s_i^s \rangle. \tag{B.28}$$

For the projection loss,

$$u_{k,i}^t = \langle s_i^t, s_i^t \rangle = 1, \qquad u_{k,i}^s = \langle s_i^t, s_i^s \rangle. \tag{B.29}$$

Since $0 \leq \langle s_i^t, s_i^s \rangle \leq 1$,

$$\mathcal{L}_{\text{proj}}^{(k)} = \big|1 - \langle s_i^t, s_i^s \rangle\big| = 1 - \langle s_i^t, s_i^s \rangle. \tag{B.30}$$

Therefore,

$$\mathcal{L}_{\text{mean}}^{(k)} = \mathcal{L}_{\text{proj}}^{(k)} = 1 - \langle \sqrt{p_i^t}, \sqrt{p_i^s} \rangle = H^2(p_i^t, p_i^s), \tag{B.31}$$

where

$$H^2(p_i^t, p_i^s) = 1 - \sum_{j=1}^{K} \sqrt{p_{ij}^t p_{ij}^s} \tag{B.32}$$

is the squared Hellinger distance under our convention. This quantity is bounded in $[0,1]$ and vanishes exactly when $p_i^t = p_i^s$, the same minimizer as KL distillation. Consequently, in the single-sample case the combined per-subset MP loss is $2H^2(p_i^t, p_i^s)$. When we refer to the single-sample squared-Hellinger endpoint, we mean that each MP term equals $H^2$; the summed loss differs only by the constant factor 2. The Hellinger distance itself is a metric; here MP recovers its squared form.

The outer square root in $\mathcal{L}_{\text{proj}}$ is essential for this exact endpoint. If the projection residual were squared without the outer root, the single-sample limit would be $H^4$ instead of $H^2$.

### B.7 Temperature effects

Temperature enters MP only through

$$p = \mathrm{softmax}(z/T), \qquad s = \sqrt{p}. \tag{B.33}$$

The square-root transformation changes the role of temperature relative to KL-based distillation. Since

$$\frac{d\sqrt{p_j}}{dp_j} = \frac{1}{2\sqrt{p_j}}, \tag{B.34}$$

small probabilities are enlarged in square-root coordinates. Thus, MP does not rely solely on a large temperature to expose low-probability coordinates. Temperature still controls the effective softened support: smaller $T$ makes predictions closer to one-hot, while larger $T$ moves them toward the simplex center.

**Proposition B.2** ($T \downarrow 0$ limit). *Assume that each teacher and student logit vector has a unique argmax, denoted $k_i^t$ and $k_i^s$. As $T \downarrow 0$,*

$$\sqrt{\mathrm{softmax}(z_i^t/T)} \to e_{k_i^t}, \qquad \sqrt{\mathrm{softmax}(z_i^s/T)} \to e_{k_i^s}. \tag{B.35}$$

*For a subset $S_k$, define the empirical teacher and student argmax histograms*

$$\pi_k^t(j) = \frac{1}{|S_k|} \sum_{i \in S_k} \mathbf{1}[k_i^t = j], \qquad \pi_k^s(j) = \frac{1}{|S_k|} \sum_{i \in S_k} \mathbf{1}[k_i^s = j]. \tag{B.36}$$

*If both histograms are nonzero, then*

$$\mathcal{L}_{\mathrm{mean}}^{(k)} \longrightarrow 1 - \frac{\langle \pi_k^t, \pi_k^s \rangle}{\|\pi_k^t\|_2 \, \|\pi_k^s\|_2}. \tag{B.37}$$

*Moreover, the teacher and student projections used by MP satisfy*

$$u_{k,i}^t \longrightarrow \frac{\pi_k^t(k_i^t)}{\|\pi_k^t\|_2}, \qquad u_{k,i}^s \longrightarrow \frac{\pi_k^t(k_i^s)}{\|\pi_k^t\|_2}. \tag{B.38}$$

*Thus, the low-temperature mean loss compares teacher and student argmax histograms at the subset level, while the projection loss compares each sample's argmax through the teacher subset histogram.*

*Proof sketch.* As $T \downarrow 0$, each softmax converges to a one-hot vector at its argmax. Hence the subset means converge to the empirical argmax histograms $\pi_k^t$ and $\pi_k^s$, and the normalized mean directions converge to

$$\frac{\pi_k^t}{\|\pi_k^t\|_2}, \qquad \frac{\pi_k^s}{\|\pi_k^s\|_2}. \tag{B.39}$$

The mean-loss limit follows immediately. For the projection term, MP projects both teacher and student predictions onto the teacher direction. Therefore,

$$(e_{k_i^t})^\top \frac{\pi_k^t}{\|\pi_k^t\|_2} = \frac{\pi_k^t(k_i^t)}{\|\pi_k^t\|_2}, \qquad (e_{k_i^s})^\top \frac{\pi_k^t}{\|\pi_k^t\|_2} = \frac{\pi_k^t(k_i^s)}{\|\pi_k^t\|_2}. \tag{B.40}$$

$\square$

### B.8 Gradient scaling

Temperature enters MP through

$$s = \sqrt{\mathrm{softmax}(z/T)}. \tag{B.41}$$

Writing $p = \mathrm{softmax}(z/T)$, we have

$$\frac{\partial p_j}{\partial z_l} = \frac{1}{T} p_j (\delta_{jl} - p_l), \qquad \frac{\partial s_j}{\partial z_l} = \frac{1}{2T} s_j (\delta_{jl} - p_l). \tag{B.42}$$

Thus the logit-to-square-root Jacobian carries an explicit $1/T$ factor. The averaging and normalization steps in the mean loss, and the rooted norm in the projection loss, do not introduce additional explicit powers of $T$; away from degenerate points, their temperature dependence is inherited through $\partial s/\partial z$. Additional decay can still occur in high-temperature regimes because teacher and student predictions both approach the uniform distribution.

For comparison, KL distillation satisfies

$$\frac{\partial D_{\mathrm{KL}}(p^t\|p^s)}{\partial z_s^l} = \frac{1}{T}(p_s^l - p_t^l). \tag{B.43}$$

At high temperature, $p_s - p_t = \mathcal{O}(1/T)$, so the KL gradient is $\mathcal{O}(1/T^2)$. This is the standard reason for multiplying KL by $T^2$. MP is first-order in square-root coordinates and has only the explicit Jacobian factor before residual-dependent cancellation. This motivates the milder $\alpha T$ scaling used in the main objective, without claiming that the MP gradient is exactly $\mathcal{O}(1/T)$ in every temperature regime.

**Per-sample versus aggregate-only matching.** Projection matching is imposed per sample rather than only through the aggregate resultant length. Let $v_i$ denote projection coordinates within one subset and compare

$$\mathcal{L}_{\mathrm{stat}} = (\bar{v}^t - \bar{v}^s)^2, \qquad \mathcal{L}_{\mathrm{full}} = \frac{1}{N}\sum_i (v_i^t - v_i^s)^2. \tag{B.44}$$

Their gradients are

$$\nabla_\theta \mathcal{L}_{\mathrm{stat}} = -\frac{2}{N}(\bar{v}^t - \bar{v}^s)\sum_i \nabla_\theta v_i^s, \qquad \nabla_\theta \mathcal{L}_{\mathrm{full}} = -\frac{2}{N}\sum_i (v_i^t - v_i^s)\nabla_\theta v_i^s. \tag{B.45}$$

The aggregate loss uses one shared residual and stops once the subset average aligns. The per-sample loss keeps one residual per prediction and continues to constrain individual samples.

## B.9 Technical details

The vMF density on $\mathbb{S}^{K-1}$ is

$$f(s \mid \mu, \kappa) = C_K(\kappa)\exp(\kappa\mu^\top s), \qquad \|\mu\|_2 = 1, \quad \kappa \geq 0, \tag{B.46}$$

with

$$C_K(\kappa) = \frac{\kappa^{K/2-1}}{(2\pi)^{K/2}I_{K/2-1}(\kappa)}. \tag{B.47}$$

Writing $\psi(\kappa) = -\log C_K(\kappa)$, the vMF is an exponential family with natural parameter $\theta = \kappa\mu$, sufficient statistic $T(s) = s$, and log-partition $\psi$. The standard identities are

$$\nabla_\theta \psi = \mathbb{E}_\theta[s], \qquad \nabla_\theta^2 \psi = \mathrm{Cov}_\theta(s) \succeq 0. \tag{B.48}$$

Using the Bessel recurrence

$$I_\nu'(x) = I_{\nu+1}(x) + \frac{\nu}{x}I_\nu(x), \qquad \nu = K/2 - 1, \tag{B.49}$$

one obtains

$$\psi'(\kappa) = \frac{I_{K/2}(\kappa)}{I_{K/2-1}(\kappa)} = A_K(\kappa), \qquad \mathbb{E}_{\mathrm{vMF}(\mu,\kappa)}[s] = A_K(\kappa)\mu. \tag{B.50}$$

For $R \in [0,1)$, the equation

$$A_K(\hat{\kappa}) = R \tag{B.51}$$

has a unique solution. In the concentrated regime,

$$\hat{\kappa} \approx \frac{K-1}{2(1-R)}, \tag{B.52}$$

while for small $R$,

$$\hat{\kappa} \approx KR\left(1 + \frac{K}{K+2}R^2 + \frac{K^2(K+8)}{(K+2)^2(K+4)}R^4\right), \tag{B.53}$$

following the standard directional-statistics approximations. MP does not use these approximations; it works directly with $R$ and per-sample projections.

The Bregman divergence generated by $\psi$ is

$$B_\psi(\kappa_s, \kappa_t) = \psi(\kappa_s) - \psi(\kappa_t) - A_K(\kappa_t)(\kappa_s - \kappa_t). \tag{B.54}$$

Expanding $D_{\mathrm{KL}}(f_t \| f_s)$ and using

$$\mathbb{E}_{f_t}[s] = A_K(\kappa_t)\mu_t \tag{B.55}$$

gives

$$D_{\mathrm{KL}}(f_t \| f_s) = A_K(\kappa_t)\kappa_s(1 - \mu_t^\top \mu_s) + B_\psi(\kappa_s, \kappa_t). \tag{B.56}$$

Finally, because square-root probabilities lie on the unit positive sphere, a fully faithful parametric model would require a density restricted to the unit positive sphere

$$p_+(s \mid \mu, \kappa) = \frac{p(s \mid \mu, \kappa)}{Z_+(\mu, \kappa)}, \qquad Z_+(\mu, \kappa) = \int_{\mathbb{S}_+^{K-1}} p(s \mid \mu, \kappa)\, ds. \tag{B.57}$$

This changes the log-partition and removes the simple closed-form relation used above. We therefore use the full-sphere vMF only to interpret the empirical statistics $(\mu, R)$; MP does not fit this model.

# C  Admissible-set geometry on the positive sphere

This appendix describes the admissible-set geometry induced by the mean-projection constraint. Given a teacher prediction $s^t = \sqrt{p^t}$ and a class-level reference direction $\mu \in \mathbb{S}_+^{K-1}$, the projection constraint $\mu^\top s = \mu^\top s^t$ defines a subset of the unit positive sphere. Measuring how this set intersects the class regions gives a class-mass vector that summarizes how the admissible set distributes over those regions. This is a geometric summary of the admissible set, not an additional training objective; we use it only to formalize the geometry illustrated in Figure 1. We state the construction for general $K$.

## C.1  Voronoi cells and the induced class-mass vector

The class regions on the unit positive sphere are determined by the largest coordinate. Equivalently, they are Voronoi cells of the simplex vertices.

**Lemma C.1** (Voronoi class regions and induced class-mass vector)**.** *Let*

$$\mathbb{S}_+^{K-1} = \{s \in \mathbb{R}_{\geq 0}^K : \|s\|_2 = 1\}. \tag{C.1}$$

*For each class $k \in \{1, \dots, K\}$, define*

$$\mathcal{R}_k = \{s \in \mathbb{S}_+^{K-1} : s_k = \max_j s_j\}. \tag{C.2}$$

*Then $\{\mathcal{R}_k\}_{k=1}^K$ partitions $\mathbb{S}_+^{K-1}$ up to measure-zero tie boundaries, and*

$$\mathcal{R}_k = \{s \in \mathbb{S}_+^{K-1} : \|s - e_k\|_2 \leq \|s - e_j\|_2 \ \forall j\}. \tag{C.3}$$

*Let $s^t \in \mathbb{S}_+^{K-1}$ be a teacher prediction and let $\mu \in \mathbb{S}_+^{K-1}$ be a reference direction. Define*

$$c = \mu^\top s^t \in [0, 1] \tag{C.4}$$

*and the positive-sphere constraint set*

$$\mathcal{M}_+(c, \mu) = \{s \in \mathbb{S}_+^{K-1} : \mu^\top s = c\}. \tag{C.5}$$

*Assume that $\mathcal{M}_+(c, \mu)$ has positive induced $(K-2)$-dimensional measure on the slice, and that class-tie boundaries have zero measure under this induced measure. Writing $\sigma$ for this induced measure, define*

$$P_k(s^t, \mu) = \frac{\sigma(\mathcal{M}_+(c, \mu) \cap \mathcal{R}_k)}{\sigma(\mathcal{M}_+(c, \mu))}, \qquad k = 1, \dots, K. \tag{C.6}$$

*Then $P(s^t, \mu) = (P_1, \dots, P_K)$ is a probability vector.*

*Proof.* For any $s \in \mathbb{S}_+^{K-1}$, at least one coordinate attains the maximum, so $s \in \bigcup_k \mathcal{R}_k$. Ties occur on lower-dimensional sets and have zero surface measure. For the Voronoi characterization,

$$\|s - e_j\|_2^2 = \|s\|_2^2 + \|e_j\|_2^2 - 2s_j = 2 - 2s_j. \tag{C.7}$$

Hence

$$\|s - e_k\|_2 \leq \|s - e_j\|_2 \quad \Longleftrightarrow \quad s_k \geq s_j. \tag{C.8}$$

Under the stated nondegeneracy condition, the sets $\mathcal{M}_+(c, \mu) \cap \mathcal{R}_k$ therefore partition $\mathcal{M}_+(c, \mu)$ up to measure-zero boundaries. Nonnegativity and normalization of $P_k$ follow immediately. $\qquad \square$

With respect to the normalized induced measure on $\mathcal{M}_+(c, \mu)$, this gives the interpretation

$$P_k(s^t, \mu) = \mathbb{P}\big(s \in \mathcal{R}_k \mid s \in \mathcal{M}_+(c, \mu)\big), \qquad c = \mu^\top s^t. \tag{C.9}$$

The teacher prediction fixes the projection level $c$, while the subset direction $\mu$ fixes the axis along which the constraint is imposed. The resulting vector describes how the admissible set is distributed across the class regions.

```
def mean_projection_loss(z_s, z_t, y, T, K, eps=1e-12):
    # Square-root transformation from simplex to unit positive sphere.
    s_s = (F.softmax(z_s / T, dim=1) + eps).sqrt()
    s_t = (F.softmax(z_t / T, dim=1) + eps).sqrt()

    # Class-induced subsets: gather samples by gold label y.
    classes, inverse = torch.unique(y, return_inverse=True)
    counts = torch.zeros(len(classes), 1, device=s_s.device)
    counts.scatter_add_(0, inverse[:, None], torch.ones_like(s_s[:, :1]))

    # Subset means of the sphere points, one row per nonempty class.
    m_s = torch.zeros(len(classes), K, device=s_s.device)
    m_t = torch.zeros(len(classes), K, device=s_t.device)
    m_s.scatter_add_(0, inverse[:, None].expand(-1, K), s_s)
    m_t.scatter_add_(0, inverse[:, None].expand(-1, K), s_t)
    m_s, m_t = m_s / counts, m_t / counts

    # Unit-norm mean directions.
    mu_s = F.normalize(m_s, p=2, dim=1, eps=eps)
    mu_t = F.normalize(m_t, p=2, dim=1, eps=eps)

    # Mean matching.
    L_mean = (1.0 - (mu_t * mu_s).sum(dim=1)).mean()

    # Projection matching along the teacher direction.
    r = (mu_t[inverse] * (s_t - s_s)).sum(dim=1)
    sq_per_class = torch.zeros(len(classes), device=s_s.device)
    sq_per_class.scatter_add_(0, inverse, r * r)
    L_proj = (sq_per_class / counts.squeeze(1)).clamp_min(eps).sqrt().mean()

    return L_mean + L_proj
```

Figure D.1: PyTorch-style pseudocode for MP with a class-wise partition. Only nonempty classes in the mini-batch are used, and the single-sample case reduces to the squared Hellinger endpoint (see Section 3.2). The displayed $\epsilon$ is numerical stabilization.

## D    Experiment settings

### D.1    Experimental design and implementation overview

This section collects the implementation details needed to reproduce the experiments in Section 4. The main text reports the primary metrics and conclusions; here we specify the training protocols, checkpoint rules, and baseline definitions.

**MP loss implementation.**    Figure D.1 gives an example PyTorch implementation of the MP loss. The function computes the square-root transformation, the class-wise subset means, the mean-matching loss, and the projection-matching loss. It returns

$$\mathcal{L}_{\mathrm{MP}} = \mathcal{L}_{\mathrm{mean}} + \mathcal{L}_{\mathrm{proj}}. \tag{D.1}$$

The trainer applies the final objective

$$\mathcal{L} = (1 - \alpha)\mathcal{L}_{\mathrm{CE}} + \alpha T \mathcal{L}_{\mathrm{MP}}, \tag{D.2}$$

as in Equation (12).

**NLI trained-student experiments.**    The experiments use the same frozen `roberta-large-mnli` teacher. We train two students: `electra-small-discriminator` and `distilbert-base-uncased`. ANLI is trained on pooled R1+R2+R3 and evaluated on each round separately; FEVER-NLI and WANLI are trained and evaluated as separate targets. All NLI students are trained for 3 epochs with AdamW, batch size 32, and three seeds. The compared methods are CE, KD, DKD, RLD, and MP. KD and MP use $\alpha = 0.9$; DKD and RLD use their published defaults here. KL baselines use $T = 4$, while MP uses $T = 2$.

**Fine-grained classification.** The fine-grained experiments in Section 4.2 use CUB-200, Stanford Cars, and Oxford-IIIT Pets. Teachers and students are ImageNet-pretrained (Russakovsky et al., 2015) transformers; the batch size is chosen so that class-wise subsets are populated without collapsing entirely to the single-sample limit.

For each dataset, we fine-tune two teachers: ViT-Base/16 (ImageNet-21k pretrained) and DINOv2-Base/14 (LVD-142M self-supervised pretrained). Teacher checkpoints are selected by minimum validation NLL with patience-5 early stopping. Both students sit at the same compact tier, DeiT-Tiny at 5M and ViT-Tiny at 5.7M parameters, but differ in pretraining recipe: DeiT-Tiny uses ImageNet-1k distillation pretraining and ViT-Tiny uses straight ImageNet-1k supervised pretraining. Each teacher is paired with both students, giving four teacher-student backbones per dataset and twelve in total.

Students are trained with AdamW, batch size 256, 50 epochs, 5-epoch linear warmup followed by cosine decay, base learning rate $10^{-4}$ (classification head at $10\times$), weight decay $5\times10^{-4}$, BF16 autocast, and three seeds. The student CE loss does not use label smoothing (label smoothing is applied only to the teacher's CE; see the next paragraph). KD and MP use a fixed $\alpha = 0.9$. DKD and RLD use the additive CE form with their published defaults (calibrated on CIFAR-100); a DKD-side $\beta$ sweep over $\beta_{\mathrm{dkd}} \in \{2, 4, 8\}$ on the 12-backbone fine-grained grid is reported in Section 4.4 (full numbers in Table 5). RLD is run at its published default throughout (no sweep, given its lower $\beta$-sensitivity on this grid). KL baselines use $T = 4$ under textbook Hinton scaling $\alpha \cdot T^2 \cdot \mathrm{KL}$ (i.e. `kd_scale=t2` in the trainer), with MP at its paper-stated $\alpha \cdot T \cdot \mathcal{L}_{\mathrm{MP}}$ weighting. MP uses $T = 2$ in the main table, with $T \in \{1, 2, 4\}$ reported as an ablation in Section 4.4.

Teacher training uses label smoothing $\varepsilon = 0.1$. This avoids near one-hot teacher predictions in the fine-grained setting. Student checkpoints are selected by validation top-1. After training, each selected checkpoint is evaluated on top-1, NLL, ECE, Brier score, and AUROC; the main text reports top-1 and ECE, and the full metrics are reported in the appendix tables.

**Office-Home distribution shift.** The Office-Home experiments in Section 4.4 use the four domains Art, Clipart, Product, and Real-World. For each source domain $S$, a teacher is fine-tuned on $S$ only, and students are distilled on the same source domain. Students are evaluated on the source test set as ID and on the union of the other three test sets as OOD. Each domain is split 80/20 per seed.

We use a single ViT-B/16 teacher (ImageNet-21k pretrained, fine-tuned per source). Teacher checkpoints are selected by minimum validation NLL with patience-5 early stopping. Each teacher is distilled into three students at two parameter tiers: DeiT-Tiny (5M) and ViT-Tiny (5.7M), and DINOv2-S/14 at 22M. Students are trained with AdamW, batch size 128, 50 epochs, 5-epoch linear warmup with cosine decay, base learning rate $10^{-4}$ (head $10\times$), weight decay 0.05, BF16 autocast, and three seeds. As in the fine-grained protocol, the student CE loss does not use label smoothing; only the teacher's CE does.

Distillation hyperparameters match the fine-grained appendix paragraph: KD and MP at $\alpha = 0.9$; DKD and RLD at their published defaults throughout. KL baselines use $T = 4$ under textbook $\alpha \cdot T^2 \cdot \mathrm{KL}$ scaling, and MP uses $T = 2$ at $\alpha \cdot T \cdot \mathcal{L}_{\mathrm{MP}}$.

We report ID/OOD ECE, OOD NLL, OOD Brier, worst-domain OOD top-1, and OOD-detection AUROC using maximum softmax probability. Per-source top-1 and extended metrics are reported in the appendix. The teacher is always trained on the same source domain used for distillation, so the OOD evaluation measures student generalization under domain shift rather than distillation from an OOD teacher.

**Per-sample mechanism analysis.** Section 4.3 probes three sample-level questions on the trained student:

- when the teacher is wrong, does the student still recover the gold label?

- when the teacher is right, does the student damage that correct prediction?

- when the teacher is wrong, does the student simply copy the teacher's wrong answer (mimicry)?

This part specifies these three metrics and how they are reported.

*Notation.* For each test example $i$, let $y_i \in \{1, \ldots, K\}$ be the gold label, $t_i$ the teacher argmax, and $s_i$ the student argmax.

*Per-sample events.* On the full test set let:

- $N_{T \neq k}$: teacher is wrong on the gold label, $\#\{i : t_i \neq y_i\}$.
- $N_{T=k}$: teacher is right on the gold label, $\#\{i : t_i = y_i\}$.
- $N_{S=k, T \neq k}$: student recovers a teacher mistake, $\#\{i : s_i = y_i, t_i \neq y_i\}$.
- $N_{S \neq k, T=k}$: student damages a teacher-correct sample, $\#\{i : s_i \neq y_i, t_i = y_i\}$.
- $N_{S=T \neq k}$: student copies a teacher mistake, $\#\{i : s_i = t_i \neq y_i\}$.

All counts are over the full test set, with no extra subset restriction.

*Reported rates.* The three rates in Section 4.3 are

$$
\begin{aligned}
\rho &= \frac{N_{S=k, T \neq k}}{N_{T \neq k}} &= \mathbb{P}(\text{student} = k \mid \text{teacher} \neq k), &\quad \text{(recovery)} \\
\delta &= \frac{N_{S \neq k, T=k}}{N_{T=k}} &= \mathbb{P}(\text{student} \neq k \mid \text{teacher} = k), &\quad \text{(damage)} \\
\omega &= \frac{N_{S=T \neq k}}{N_{T \neq k}} &= \mathbb{P}(\text{student} = \text{teacher} \mid \text{teacher} \neq k). &\quad \text{(mimicry)}
\end{aligned}
$$

Higher $\rho$ and lower $\delta$, $\omega$ favor the student. Each test example contributes to either the teacher-wrong partition (used by $\rho$ and $\omega$) or the teacher-right partition (used by $\delta$).

These rates are computed per seed and reported as mean ± std across the three student seeds for each (dataset, teacher, student) backbone. Cross-backbone summaries (e.g. the mean row in Table 3) average each per-backbone mean over the fine-grained backbones.

**CIFAR-100 single-sample check.** The CIFAR-100 experiment verifies the single-sample endpoint of MP. When $|S_k| = 1$, each MP term reduces to the squared Hellinger distance and the summed MP loss is the constant multiple $2H^2$. We therefore run the equivalent pointwise Hellinger objective as a standalone distillation loss under the standard CRD/mdistiller CIFAR-100 from-scratch protocol (Tian et al., 2020; Zhao et al., 2022). The purpose of this check is to evaluate whether the bounded pointwise endpoint remains comparable to KD in a conventional evaluation protocol.

We follow the standard CIFAR-100 setup: 240 epochs, SGD with momentum 0.9, cosine schedule, weight decay $5 \times 10^{-4}$, batch size 64, and three seeds. Both teacher and student are trained from scratch on CIFAR-100 under this protocol; the mdistiller reference (Zhao et al., 2022) supplies the protocol and the architecture definitions. We evaluate the teacher-student pairs reported in Table E.12. KL baselines use $T = 4$; the Hellinger endpoint uses $T = 2$, following the MP temperature convention.

**Checkpoint selection and reporting.** Teacher checkpoints in fine-grained classification and Office-Home are selected by validation NLL, because the teacher provides a soft distributional target. Student checkpoints are selected by validation top-1, and calibration metrics are computed post hoc from the selected checkpoints. Unless otherwise stated, reported numbers are means over three seeds. Office-Home additionally averages over the four source-domain rotations.

## D.2 Baselines

We compare MP with three representative logit-side distillation methods: standard knowledge distillation (KD) (Hinton et al., 2015), Decoupled Knowledge Distillation (DKD) (Zhao et al., 2022), and Refined Logit Distillation (RLD) (Sun et al., 2025). All three operate on teacher and student output distributions, but differ in how they weight, decompose, or mask the teacher signal. The comparison is intentionally restricted to output-probability distillation. Feature-level distillation methods are complementary but not direct competitors because they require hidden representations or attention maps, while MP uses only teacher

logits, student logits, and labels. Other logit-side variants such as MLKD, CTKD, and LSKD are discussed in Sections 2.1 and 2.2.

Temperature and loss weights are protocol choices. The KL baselines use the standard temperature-scaled convention $T^2 D_{\mathrm{KL}}$ with $T = 4$, and DKD/RLD use their published weights. MP uses the $T\mathcal{L}_{\mathrm{MP}}$ scaling motivated by the square-root Jacobian in Section B.8. Its setting $T = 2$ is further discussed in Table E.8.

*KD and MP* use a standard KD-style mixing,

$$\mathcal{L} = (1 - \alpha)\,\mathcal{L}_{\mathrm{CE}} + \alpha\,c_T\,\mathcal{L}_{\mathrm{method}}, \qquad \alpha = 0.9, \tag{D.3}$$

where

$$\mathcal{L}_{\mathrm{CE}} = -\log q_{i,y_i} \tag{D.4}$$

with $q_i = \mathrm{softmax}(z_i^s)$, and $c_T$ is a temperature prefactor ($c_T = 1$ for KD because the $T^2$ factor is already inside $\mathcal{L}_{\mathrm{KD}}$; $c_T = T$ for MP).

*DKD and RLD* use the additive form from their original papers,

$$\mathcal{L} = \mathcal{L}_{\mathrm{CE}} + \mathcal{L}_{\mathrm{method}}, \tag{D.5}$$

where the cross-entropy weight is fixed at one and the distillation term $\mathcal{L}_{\mathrm{method}}$ carries its own internal coefficients $\alpha_{\mathrm{method}}$ and $\beta_{\mathrm{method}}$. CE is always present in the total training loss.

### D.2.1  Hinton Knowledge Distillation

For temperature-scaled teacher and student distributions

$$p_i^t = \mathrm{softmax}\left(\frac{z_i^t}{T}\right), \qquad p_i^s = \mathrm{softmax}\left(\frac{z_i^s}{T}\right), \tag{D.6}$$

the KD method loss is the temperature-scaled KL divergence

$$\mathcal{L}_{\mathrm{KD}} = T^2\,D_{\mathrm{KL}}\big(p_i^t \,\|\, p_i^s\big). \tag{D.7}$$

The final objective follows (D.3) with $c_T = 1$ (the $T^2$ factor is already inside $\mathcal{L}_{\mathrm{KD}}$). Defaults: $\alpha{=}0.9$, $T{=}4$.

### D.2.2  Decoupled Knowledge Distillation

DKD (Zhao et al., 2022) decomposes the distillation signal into a target-class (TCKD) and non-target-class (NCKD) term:

$$\mathcal{L}_{\mathrm{DKD}} = T^2\big(\alpha_{\mathrm{dkd}}\,\mathcal{L}_{\mathrm{TCKD}} + \beta_{\mathrm{dkd}}\,\mathcal{L}_{\mathrm{NCKD}}\big), \tag{D.8}$$

where the $T^2$ prefactor is inherited from the standard Hinton temperature scaling. Defaults: the DKD-paper values calibrated on CIFAR-100, with $T{=}4$. We additionally sweep $\beta_{\mathrm{dkd}} \in \{2, 4, 8\}$ on the fine-grained grid (Table 5).

### D.2.3  Refined Logit Distillation

RLD (Sun et al., 2025) refines the teacher signal with a sample-confidence term (SCD) and a masked class-distribution term (MCD):

$$\mathcal{L}_{\mathrm{RLD}} = T^2\big(\alpha_{\mathrm{rld}}\,\mathcal{L}_{\mathrm{SCD}} + \beta_{\mathrm{rld}}\,\mathcal{L}_{\mathrm{MCD}}\big), \tag{D.9}$$

with both terms computed at $T{=}4$. Following (D.5), the training loss is $\mathcal{L} = \mathcal{L}_{\mathrm{CE}} + \mathcal{L}_{\mathrm{RLD}}$ (CE included), matching the additive form of the RLD paper. RLD is run at its published default throughout (no $\beta$ sweep).

### D.2.4  Mean Projection

MP uses the loss defined in Section 3:

$$\mathcal{L}_{\mathrm{MP}} = \mathcal{L}_{\mathrm{mean}} + \mathcal{L}_{\mathrm{proj}}. \tag{D.10}$$

The total loss following (D.3) at $c_T = T$ (paper-stated $\alpha{\cdot}T{\cdot}\mathcal{L}_{\mathrm{MP}}$ weighting). Defaults: $\alpha{=}0.9$, $T{=}2$.

Table E.1: Office-Home distribution shift. Domain-average columns aggregate over four source domains and three seeds ($n = 12$); worst-OOD source columns are evaluated on the source with the lowest OOD top-1, which is Product for all student-method pairs, and report mean ± std over three seeds. OOD−ID NLL is the domain-average OOD NLL minus ID NLL, with lower values indicating better robustness. **Bold** marks the best distillation result per student and column. Extended metrics are reported in Table E.4.

| Student | Method | Domain average | | Worst-OOD source (Product) | | Robustness |
|---|---|---|---|---|---|---|
| | | ECE↓ | OOD Brier↓ | ECE↓ | OOD Brier↓ | OOD−ID NLL↓ |
| DeiT-Tiny | CE | 22.71 ±3.22 | 0.666 ±0.046 | 25.29 ±1.01 | 0.723 ±0.019 | 1.492 |
| | KD | 4.05 ±1.43 | 0.588 ±0.033 | 5.46 ±0.95 | 0.639 ±0.002 | 1.202 |
| | DKD | 6.37 ±2.46 | 0.590 ±0.032 | 8.32 ±1.13 | 0.638 ±0.015 | 1.238 |
| | RLD | 7.00 ±2.66 | 0.594 ±0.033 | 9.38 ±1.09 | 0.644 ±0.012 | 1.239 |
| | MP | **3.66** ±0.93 | **0.580** ±0.032 | **4.36** ±0.30 | **0.629** ±0.010 | **1.160** |
| ViT-Tiny | CE | 24.71 ±3.42 | 0.709 ±0.039 | 27.66 ±0.46 | 0.762 ±0.006 | 1.762 |
| | KD | 4.49 ±1.32 | 0.601 ±0.038 | **3.48** ±0.40 | 0.650 ±0.012 | 1.269 |
| | DKD | 4.22 ±1.17 | 0.626 ±0.059 | 4.73 ±0.73 | 0.684 ±0.018 | 1.371 |
| | RLD | **4.12** ±1.25 | 0.604 ±0.044 | 5.28 ±1.13 | 0.656 ±0.017 | 1.313 |
| | MP | 6.31 ±3.65 | **0.593** ±0.029 | 4.07 ±0.79 | **0.625** ±0.011 | **1.213** |
| DINOv2-S/14 | CE | 22.35 ±12.28 | 0.697 ±0.122 | 35.07 ±0.82 | 0.857 ±0.020 | 1.934 |
| | KD | **4.46** ±1.35 | 0.548 ±0.045 | **3.70** ±0.30 | 0.613 ±0.013 | **1.280** |
| | DKD | 5.18 ±3.03 | 0.554 ±0.056 | 5.44 ±1.06 | **0.604** ±0.010 | 1.324 |
| | RLD | 5.10 ±2.53 | **0.547** ±0.051 | 6.41 ±0.67 | 0.611 ±0.014 | 1.304 |
| | MP | 5.73 ±3.03 | 0.578 ±0.057 | 9.38 ±0.76 | 0.660 ±0.017 | 1.321 |

# E   Extended experiments and analyses

## E.1   Office-Home distribution shift: full results

Section 4.4 reports the OOD−ID NLL gap as a single robustness summary. This subsection gives the full Office-Home results: setup details, the full main calibration table (OOD ECE, OOD Brier on both domain-average and worst-source views, Table E.1), the per-source ID/OOD top-1 and ECE breakdown (Tables E.2 and E.3), and source-averaged extended metrics (ID NLL, OOD NLL, OOD−ID NLL gap, ID Brier, worst-domain OOD top-1) in Table E.4.

We use Office-Home (Venkateswara et al., 2017), which contains four visually distinct domains (*Art*, *Clipart*, *Product*, *Real-World*) and $K = 65$ classes. For each source domain $S$, we fine-tune a ViT-B/16 teacher on $S$, distill students on the same source domain, and evaluate on both the source test split (ID) and the union of the other three test splits (OOD). Each domain rotates through the source role, giving four source-domain settings per student and method.

We distill the teacher into three students: DeiT-Tiny, ViT-Tiny, and DINOv2-S/14. DeiT-Tiny and ViT-Tiny have similar model size but different pretraining schemes, while DINOv2-S/14 tests a larger student. Teacher checkpoints are selected by validation NLL and student checkpoints by validation top-1. Students are trained with the same 50-epoch AdamW fine-tuning strategy as in Section 4.2, with batch size 128; full details are given in Section D.1.

We compare MP with CE, KD, DKD, and RLD under the same scaling conventions as in Section 4.2: KL baselines use $T^2$ scaling and $T = 4$, while MP uses $T$ weighting and $T = 2$. KD and MP use $\alpha = 0.9$; DKD and RLD use their published defaults.

Table E.1 shows that all distillation methods substantially reduce OOD ECE relative to CE, whose OOD ECE remains above 20% for all three students. Among the distillation methods, MP is strongest on DeiT-Tiny: it gives the best domain-average ECE and OOD Brier, the best worst-source ECE and OOD Brier,

Table E.2: Office-Home per-source breakdown (part 1 / 2: Art and Clipart sources). Each source domain is used for teacher fine-tuning and student distillation; OOD evaluation uses the union of the other three domains. Teacher is ViT-B/16 (IN-21k). KD/MP at $\alpha$=0.9; DKD and RLD at their published defaults. Results are mean ± std over three seeds. Sources Product and Real-World are reported in Table E.3.

| Source | Student | Method | Top-1 (%) | | ECE (%) | |
| | | | ID↑ | OOD↑ | ID↓ | OOD↓ |
|---|---|---|---|---|---|---|
| Art | DeiT-Tiny | CE | 73.75 ±1.33 | 53.81 ±0.51 | 12.67 ±0.87 | 21.91 ±1.28 |
| | | KD | 76.98 ±1.14 | 54.98 ±1.43 | 10.38 ±2.29 | **2.48** ±0.92 |
| | | DKD | 76.70 ±2.63 | **56.16** ±1.59 | 8.91 ±1.95 | 3.54 ±0.65 |
| | | RLD | 77.11 ±1.80 | 55.89 ±1.20 | **6.99** ±1.75 | 3.62 ±0.67 |
| | | MP | **77.66** ±2.56 | 55.37 ±0.78 | 10.62 ±2.98 | 3.07 ±0.35 |
| | ViT-Tiny | CE | 72.30 ±1.21 | 50.62 ±0.49 | 12.02 ±1.67 | 23.32 ±0.70 |
| | | KD | 78.01 ±1.49 | **55.03** ±0.98 | 12.86 ±2.11 | 4.45 ±0.30 |
| | | DKD | 78.35 ±1.65 | 54.20 ±0.65 | 11.41 ±1.43 | 2.82 ±0.53 |
| | | RLD | **78.49** ±1.02 | 54.71 ±0.60 | **9.61** ±1.48 | **2.60** ±0.60 |
| | | MP | 76.49 ±1.44 | 53.98 ±0.99 | 13.18 ±2.61 | 5.51 ±1.07 |
| | DINOv2-S/14 | CE | 77.59 ±2.19 | 56.50 ±2.71 | 8.86 ±5.59 | 12.21 ±13.94 |
| | | KD | 84.95 ±2.32 | **60.92** ±0.30 | 13.60 ±2.71 | 4.86 ±0.77 |
| | | DKD | **85.09** ±3.10 | 60.75 ±0.92 | 10.59 ±3.10 | 2.52 ±0.48 |
| | | RLD | 84.95 ±3.32 | 60.60 ±0.93 | **9.12** ±3.07 | **2.51** ±0.32 |
| | | MP | 81.72 ±0.93 | 58.95 ±0.74 | 9.88 ±1.57 | 2.75 ±0.90 |
| Clipart | DeiT-Tiny | CE | 80.95 ±0.86 | 57.83 ±1.04 | 9.93 ±1.24 | 18.19 ±1.12 |
| | | KD | **83.47** ±0.97 | 58.72 ±0.49 | 6.94 ±2.16 | 4.86 ±0.75 |
| | | DKD | 83.43 ±0.43 | 57.86 ±1.25 | 6.07 ±0.90 | 8.07 ±2.91 |
| | | RLD | 82.82 ±0.41 | 58.20 ±0.83 | **6.02** ±1.57 | 8.14 ±2.95 |
| | | MP | 82.21 ±0.37 | **59.52** ±0.83 | 6.57 ±0.79 | **4.14** ±1.04 |
| | ViT-Tiny | CE | 76.10 ±1.28 | 53.51 ±2.43 | 12.09 ±0.84 | 20.07 ±0.89 |
| | | KD | **79.38** ±0.50 | 53.77 ±1.87 | 10.01 ±2.36 | 6.16 ±1.70 |
| | | DKD | 77.66 ±1.59 | 48.28 ±1.73 | 7.24 ±2.37 | 5.44 ±0.34 |
| | | RLD | 79.15 ±0.72 | 52.60 ±2.47 | **6.75** ±0.96 | **4.81** ±0.18 |
| | | MP | 77.74 ±0.63 | **56.30** ±2.36 | 10.74 ±1.14 | 12.12 ±0.65 |
| | DINOv2-S/14 | CE | 83.20 ±0.87 | 51.74 ±0.64 | 10.41 ±0.81 | 26.67 ±1.48 |
| | | KD | **85.99** ±1.03 | **59.27** ±1.32 | 6.61 ±0.63 | **5.99** ±1.30 |
| | | DKD | 84.92 ±1.61 | 55.05 ±1.76 | 6.14 ±0.89 | 9.36 ±2.64 |
| | | RLD | 85.15 ±1.06 | 58.24 ±2.09 | **5.63** ±0.63 | 8.15 ±1.79 |
| | | MP | 84.84 ±0.89 | 57.19 ±2.02 | 6.20 ±1.16 | 7.44 ±1.44 |

and the smallest OOD−ID NLL gap. On ViT-Tiny, the methods are more closely matched. MP gives the smallest OOD−ID NLL gap and remains competitive on the Product worst-source metrics, while DKD, RLD, and KD each lead one of the other calibration or Brier columns. On the larger DINOv2-S/14 student, KD gives the best domain-average ECE, average Brier, worst-source ECE, and OOD−ID NLL gap, while DKD gives the best worst-source Brier. Thus, under Office-Home shift, MP is most effective for DeiT-Tiny and remains competitive on ViT-Tiny, whereas KD is stronger for the higher-capacity DINOv2-S/14 student.

## E.2 Fine-grained classification: per-dataset and overall means

Table E.5 reports the per-dataset and overall means of top-1 and ECE for each method, averaged from Table 2 (four backbones per dataset). The three dataset means average to the *All twelve* row, which is

Table E.3: Office-Home per-source breakdown (part 2 / 2: Product and Real-World sources). Setup and conventions are as in Table E.2; sources Art and Clipart are reported there.

| Source | Student | Method | Top-1 (%) | | ECE (%) | |
|---|---|---|---|---|---|---|
| | | | ID↑ | OOD↑ | ID↓ | OOD↓ |
| Product | DeiT-Tiny | CE | 92.49 ±1.60 | 50.79 ±1.40 | 3.60 ±0.93 | 25.29 ±1.01 |
| | | KD | 92.57 ±0.70 | 52.49 ±0.29 | 9.09 ±1.02 | 5.46 ±0.95 |
| | | DKD | 92.79 ±0.49 | 52.61 ±0.80 | 5.43 ±0.89 | 8.32 ±1.13 |
| | | RLD | **93.17** ±0.80 | 52.76 ±0.45 | **5.39** ±0.74 | 9.38 ±1.09 |
| | | MP | 92.57 ±0.92 | **52.85** ±1.19 | 9.26 ±1.50 | **4.36** ±0.30 |
| | ViT-Tiny | CE | 92.08 ±1.47 | 48.50 ±0.78 | 4.00 ±1.47 | 27.66 ±0.46 |
| | | KD | 92.53 ±0.28 | 50.14 ±1.11 | 10.91 ±1.30 | **3.48** ±0.40 |
| | | DKD | 92.08 ±0.26 | 46.90 ±1.57 | 8.33 ±1.26 | 4.73 ±0.73 |
| | | RLD | **92.76** ±0.92 | 49.60 ±1.85 | **6.82** ±1.32 | 5.28 ±1.13 |
| | | MP | 92.42 ±0.56 | **52.01** ±0.81 | 11.27 ±0.61 | 4.07 ±0.79 |
| | DINOv2-S/14 | CE | 91.74 ±0.07 | 45.58 ±1.41 | 4.81 ±0.36 | 35.07 ±0.82 |
| | | KD | 94.74 ±0.34 | 53.73 ±1.49 | 8.98 ±1.00 | **3.70** ±0.30 |
| | | DKD | 94.97 ±0.36 | **54.28** ±0.27 | 5.80 ±1.04 | 5.44 ±1.06 |
| | | RLD | **95.31** ±0.26 | 54.06 ±1.06 | **5.71** ±0.57 | 6.41 ±0.67 |
| | | MP | 93.54 ±0.72 | 50.56 ±1.36 | 7.41 ±0.38 | 9.38 ±0.76 |
| Real-World | DeiT-Tiny | CE | 83.54 ±0.76 | 56.14 ±0.19 | 8.96 ±0.31 | 25.45 ±0.40 |
| | | KD | 85.23 ±0.63 | 58.21 ±0.43 | 10.03 ±1.01 | 3.39 ±0.81 |
| | | DKD | 85.15 ±0.18 | 57.69 ±0.18 | 6.39 ±0.38 | 5.56 ±0.27 |
| | | RLD | **85.34** ±0.46 | 57.23 ±0.25 | **6.28** ±0.90 | 6.87 ±0.90 |
| | | MP | 85.15 ±0.18 | **58.33** ±0.51 | 9.74 ±1.00 | **3.05** ±1.14 |
| | ViT-Tiny | CE | 85.34 ±0.33 | 54.39 ±0.07 | 7.82 ±0.68 | 27.81 ±0.28 |
| | | KD | 88.56 ±1.72 | 57.94 ±1.10 | 13.78 ±1.05 | 3.86 ±0.41 |
| | | DKD | 88.10 ±1.94 | 57.85 ±0.98 | 10.86 ±0.64 | 3.87 ±0.92 |
| | | RLD | **88.79** ±1.45 | **57.97** ±0.39 | **9.52** ±0.16 | 3.80 ±0.76 |
| | | MP | 88.17 ±0.75 | 57.51 ±1.28 | 12.84 ±1.02 | **3.53** ±0.56 |
| | DINOv2-S/14 | CE | 84.77 ±0.33 | 57.84 ±1.58 | 5.42 ±3.48 | 15.46 ±11.68 |
| | | KD | 90.28 ±0.83 | 62.75 ±0.82 | 11.87 ±0.46 | **3.29** ±0.97 |
| | | DKD | **90.89** ±0.48 | **63.67** ±1.06 | 7.95 ±0.20 | 3.38 ±0.54 |
| | | RLD | 90.05 ±1.29 | 63.28 ±0.67 | **6.73** ±0.54 | 3.36 ±0.51 |
| | | MP | 88.29 ±1.44 | 61.25 ±0.67 | 9.97 ±1.51 | 3.34 ±0.90 |

the source of the summary in Section 4.2: averaged over all settings MP attains the lowest ECE among distillation methods, while its top-1 is on par with the strongest baseline.

### E.3   Per-sample recovery analysis

This subsection reports the full per-sample diagnostic on all fine-grained backbones (three datasets × two teachers × two students) under the protocol of Section 4.2. The rates $\rho$, $\delta$, and $\omega$ follow the definitions in Section 4.3.

### E.4   MP ablations

This subsection reports the full numbers for the MP ablation summaries in Section 4.4. The MP ablations (temperature, batch size, subset choice, and loss components) use the two backbones CUB-200 / ViT-B/16 →

Table E.4: Office-Home source-averaged extended metrics, complementing Table E.1. ID NLL, OOD NLL, the OOD−ID NLL gap, ID Brier, worst-OOD top-1, and MSP-AUROC. KD/MP at $\alpha$=0.9; DKD and RLD at their published defaults. Mean ± std over four sources and three seeds (n=12). The four distillation methods are within ∼0.01 on MSP-AUROC across all three students, so the main table omits this column.

| Student | Method | ID NLL↓ | OOD NLL↓ | OOD−ID NLL↓ | ID Brier↓ | worst-OOD↑ | MSP-AUROC↑ |
|---|---|---|---|---|---|---|---|
| | CE | 0.799 ±0.335 | 2.291 ±0.257 | 1.492 | 0.271 ±0.109 | 41.21 ±6.05 | 0.720 ±0.073 |
| | KD | 0.683 ±0.227 | 1.885 ±0.153 | 1.202 | 0.239 ±0.083 | 42.34 ±6.50 | 0.711 ±0.067 |
| DeiT-Tiny | DKD | **0.654** ±0.243 | 1.892 ±0.158 | 1.238 | **0.234** ±0.088 | 42.41 ±5.99 | 0.718 ±0.070 |
| | RLD | 0.658 ±0.239 | 1.896 ±0.157 | 1.239 | 0.235 ±0.087 | 42.46 ±5.78 | 0.716 ±0.070 |
| | MP | 0.693 ±0.226 | **1.853** ±0.143 | **1.160** | 0.243 ±0.082 | **42.47** ±6.06 | **0.719** ±0.067 |
| | CE | 0.903 ±0.424 | 2.666 ±0.286 | 1.762 | 0.286 ±0.121 | 34.40 ±6.74 | 0.718 ±0.071 |
| | KD | 0.719 ±0.254 | 1.988 ±0.171 | 1.269 | 0.248 ±0.086 | **38.13** ±5.13 | 0.723 ±0.066 |
| ViT-Tiny | DKD | 0.719 ±0.273 | 2.090 ±0.266 | 1.371 | 0.250 ±0.091 | 36.23 ±3.97 | **0.729** ±0.070 |
| | RLD | **0.686** ±0.272 | 1.999 ±0.195 | 1.313 | **0.239** ±0.092 | 37.43 ±4.16 | 0.728 ±0.070 |
| | MP | 0.749 ±0.290 | **1.962** ±0.123 | **1.213** | 0.257 ±0.098 | 37.94 ±6.24 | 0.728 ±0.068 |
| | CE | 0.758 ±0.303 | 2.692 ±0.838 | 1.934 | 0.243 ±0.084 | 38.86 ±4.10 | 0.729 ±0.075 |
| | KD | 0.522 ±0.169 | **1.803** ±0.213 | **1.280** | 0.181 ±0.063 | **44.74** ±4.26 | 0.722 ±0.058 |
| DINOv2-S/14 | DKD | 0.504 ±0.197 | 1.828 ±0.250 | 1.324 | 0.179 ±0.070 | 44.00 ±3.18 | **0.730** ±0.061 |
| | RLD | **0.503** ±0.195 | 1.808 ±0.228 | 1.304 | **0.178** ±0.070 | 44.06 ±3.56 | 0.730 ±0.058 |
| | MP | 0.583 ±0.181 | 1.904 ±0.240 | 1.321 | 0.200 ±0.066 | 42.64 ±4.25 | 0.722 ±0.059 |

Table E.5: Mean top-1 and ECE (%) by dataset and over all twelve (dataset, teacher, student) settings, averaged from Table 2; each per-dataset value averages the four backbones, and the three dataset means average to the *All twelve* row. **Bold** marks the best distillation method per row.

| | CE | KD | DKD | RLD | MP |
|---|---|---|---|---|---|
| *top-1 (%)*↑ | | | | | |
| CUB-200 | 77.93 | 80.01 | 79.72 | **80.72** | 80.05 |
| Stanford Cars | 70.52 | 72.82 | 70.45 | 73.81 | **74.45** |
| Oxford Pets | 90.41 | 91.23 | 91.29 | **91.45** | 91.11 |
| All twelve | 79.62 | 81.36 | 80.49 | **81.99** | 81.87 |
| *ECE (%)*↓ | | | | | |
| CUB-200 | 7.82 | 20.49 | 29.08 | 21.33 | **13.61** |
| Stanford Cars | 10.71 | 28.28 | 38.11 | 29.88 | **17.18** |
| Oxford Pets | 1.89 | 9.90 | 11.42 | **8.29** | 8.67 |
| All twelve | 6.80 | 19.56 | 26.20 | 19.83 | **13.15** |

ViT-Tiny and Stanford Cars / ViT-B/16 → ViT-Tiny, with all other hyperparameters fixed to the protocol in Section 4.2.

**MP temperature.** Table E.8 sweeps $T \in \{1, 2, 4\}$ for MP at fixed $\alpha = 0.9$. Lower temperature improves ECE while top-1 peaks at $T = 2$, an accuracy-calibration trade-off that motivates the default $T = 2$ used in the main experiments.

**Batch size (subset size).** Table E.9 reports the batch-size sweep under two complementary protocols. The *same-updates* protocol fixes the optimizer-step count so that larger $B$ sees more data; the *fixed-epochs* protocol fixes the data exposure so that larger $B$ receives fewer updates. These protocols answer different questions and should not be compared row-wise across the two blocks.

Table E.6: Per-sample recovery analysis (part 1 / 2: top-1 and recovery $\rho$) on the full 12-backbone fine-grained grid.

| | top-1 (%)↑ | | | $\rho$ (%)↑ | | |
|---|---|---|---|---|---|---|
| Backbone | KD | MP | $\Delta$ | KD | MP | $\Delta$ |
| *CUB / ViT-B → DeiT-T* | 77.79 | 77.83 | +0.04 | 26.70 | 30.88 | +4.18 |
| CUB / ViT-B → ViT-T | 81.75 | 82.44 | +0.69 | 30.54 | 33.59 | +3.05 |
| CUB / DINOv2-B → DeiT-T | 78.45 | 77.72 | −0.73 | 26.08 | 26.03 | −0.05 |
| CUB / DINOv2-B → ViT-T | 82.06 | 82.23 | +0.17 | 27.03 | 28.38 | +1.35 |
| *Cars / ViT-B → DeiT-T* | 75.03 | 75.62 | +0.59 | 31.34 | 33.70 | +2.36 |
| Cars / ViT-B → ViT-T | 71.88 | 73.61 | +1.73 | 28.09 | 29.83 | +1.74 |
| Cars / DINOv2-B → DeiT-T | 74.42 | 75.37 | +0.95 | 22.84 | 24.63 | +1.79 |
| Cars / DINOv2-B → ViT-T | 69.97 | 73.19 | +3.22 | 20.10 | 22.89 | +2.79 |
| Pets / ViT-B → DeiT-T | 91.54 | 91.47 | −0.07 | 30.98 | 34.85 | +3.87 |
| Pets / ViT-B → ViT-T | 90.91 | 90.71 | −0.20 | 29.88 | 35.13 | +5.25 |
| *Pets / DINOv2-B → DeiT-T* | 91.78 | 91.62 | −0.16 | 27.56 | 32.59 | +5.03 |
| Pets / DINOv2-B → ViT-T | 90.70 | 90.62 | −0.08 | 27.75 | 31.66 | +3.91 |
| mean over 12 backbones | 81.36 | 81.87 | +0.51 | 27.41 | 30.35 | +2.94 |

Table E.7: Per-sample recovery analysis (part 2 / 2: damage $\delta$ and mimicry $\omega$) on the full 12-backbone fine-grained grid.

| | $\delta$ (%)↓ | | | $\omega$ (%)↓ | | |
|---|---|---|---|---|---|---|
| Backbone | KD | MP | $\Delta$ | KD | MP | $\Delta$ |
| *CUB / ViT-B → DeiT-T* | 12.87 | 13.59 | +0.72 | 39.55 | 35.42 | −4.13 |
| CUB / ViT-B → ViT-T | 8.89 | 8.64 | −0.25 | 39.74 | 37.17 | −2.57 |
| CUB / DINOv2-B → DeiT-T | 14.74 | 15.57 | +0.83 | 39.54 | 36.64 | −2.90 |
| CUB / DINOv2-B → ViT-T | 10.80 | 10.78 | −0.02 | 43.24 | 41.84 | −1.40 |
| *Cars / ViT-B → DeiT-T* | 15.46 | 15.26 | −0.20 | 28.46 | 24.70 | −3.76 |
| Cars / ViT-B → ViT-T | 18.59 | 16.86 | −1.73 | 30.20 | 27.79 | −2.41 |
| Cars / DINOv2-B → DeiT-T | 20.89 | 20.02 | −0.87 | 41.44 | 39.40 | −2.04 |
| Cars / DINOv2-B → ViT-T | 25.50 | 22.24 | −3.26 | 42.64 | 38.61 | −4.03 |
| Pets / ViT-B → DeiT-T | 4.20 | 4.55 | +0.35 | 56.43 | 50.62 | −5.81 |
| Pets / ViT-B → ViT-T | 4.80 | 5.39 | +0.59 | 56.57 | 49.52 | −7.05 |
| *Pets / DINOv2-B → DeiT-T* | 4.93 | 5.35 | +0.42 | 59.40 | 51.40 | −8.00 |
| Pets / DINOv2-B → ViT-T | 6.07 | 6.36 | +0.29 | 58.29 | 52.89 | −5.40 |
| mean over 12 backbones | 12.31 | 12.05 | −0.26 | 44.62 | 40.50 | −4.12 |

**Subset choice.** Table E.10 compares the class-wise partition with three alternatives: teacher-argmax grouping (subsets defined by the teacher's argmax), batch-as-one (the entire mini-batch as a single subset), and batch-mean-only (batch-level mean matching only). Class-wise and teacher-argmax grouping perform almost identically here, since the teacher argmax usually agrees with the gold label on these backbones; both clearly outperform the batch-level variants in top-1.

**Loss components.** Table E.11 compares the full MP objective with its two single-loss variants, which keep only $\mathcal{L}_{\text{mean}}$ or only $\mathcal{L}_{\text{proj}}$. The full loss gives the best accuracy-calibration tradeoff.

Table E.8: MP temperature ablation on two fine-grained backbones (ViT-B/16 $\rightarrow$ ViT-Tiny). All rows are MP; only $T$ varies. Mean $\pm$ std over three seeds.

| | CUB-200 | | Stanford Cars | |
|---|---|---|---|---|
| $T$ | T1$\uparrow$ | ECE$\downarrow$ | T1$\uparrow$ | ECE$\downarrow$ |
| 1 | 81.20 $\pm$0.09 | **10.36** $\pm$0.56 | 71.43 $\pm$0.55 | **8.55** $\pm$0.40 |
| 2 | **82.48** $\pm$0.17 | 15.04 $\pm$0.19 | **73.61** $\pm$0.46 | 18.13 $\pm$0.19 |
| 4 | 82.27 $\pm$0.36 | 13.97 $\pm$0.65 | 72.46 $\pm$0.68 | 19.14 $\pm$0.28 |

Table E.9: Subset-size ablation under two protocols on two fine-grained backbones (ViT-B/16 $\rightarrow$ ViT-Tiny). $\mathbb{E}|S_k| \approx B/K$. Mean $\pm$ std over three seeds.

| | | Same-updates | | | | Fixed-epochs | | | |
|---|---|---|---|---|---|---|---|---|---|
| | | CUB-200 | | Stanford Cars | | CUB-200 | | Stanford Cars | |
| $B$ | $\mathbb{E}|S_k|$ | T1$\uparrow$ | ECE$\downarrow$ | T1$\uparrow$ | ECE$\downarrow$ | T1$\uparrow$ | ECE$\downarrow$ | T1$\uparrow$ | ECE$\downarrow$ |
| 64 | 0.3 | 80.20 | 17.99 | 50.39 | 24.76 | **83.67** | **13.40** | **79.50** | **16.50** |
| 128 | 0.6 | 81.79 | 16.40 | 60.73 | 24.23 | 83.19 | 13.99 | 77.24 | 17.05 |
| 256 | 1.3 | **82.24** | 15.24 | 67.57 | 20.97 | 82.45 | 14.85 | 73.66 | 18.11 |
| 512 | 2.6 | 82.22 | 14.61 | 70.93 | 18.45 | 80.56 | 16.30 | 65.93 | 21.05 |
| 1024 | 5.2 | 81.94 | **14.22** | **73.05** | **16.97** | 73.66 | 24.53 | 45.81 | 22.57 |

**Single-sample endpoint on CIFAR-100.** Table E.12 reports the single-sample Hellinger endpoint on CIFAR-100 from scratch under the CRD/`mdistiller` protocol (Tian et al., 2020). It stays within half a point of KD in top-1 and is comparable in calibration.

**DKD non-target coefficient on Pets.** Table E.13 reports the DKD $\beta$-sweep on the Oxford-IIIT Pets backbones; Table 5 in the main paper reports the harder CUB-200 and Stanford Cars blocks. The effect of $\beta_{\mathrm{dkd}}$ is smaller on Pets.

Table E.10: Subset-choice ablation on two fine-grained backbones (ViT-B/16 $\rightarrow$ ViT-Tiny). The class-wise (gold-label) partition is the default used in the main paper. Mean $\pm$ std over three seeds.

| | CUB-200 | | Stanford Cars | |
|---|---|---|---|---|
| Subset mode | T1$\uparrow$ | ECE$\downarrow$ | T1$\uparrow$ | ECE$\downarrow$ |
| Class-induced (default) | 82.47 $\pm$0.18 | 15.03 $\pm$0.19 | **73.64** $\pm$0.53 | 18.11 $\pm$0.16 |
| Teacher-argmax | **82.48** $\pm$0.17 | 15.03 $\pm$0.18 | 73.63 $\pm$0.46 | 18.08 $\pm$0.14 |
| Batch-as-one | 78.98 $\pm$0.30 | 16.72 $\pm$0.12 | 65.69 $\pm$0.49 | 19.73 $\pm$0.53 |
| Batch-mean-only | 79.76 $\pm$0.30 | **8.29** $\pm$2.76 | 67.48 $\pm$0.67 | **9.20** $\pm$0.58 |

Table E.11: Loss-component ablation on two fine-grained backbones (ViT-B/16 → ViT-Tiny). KD is reproduced from Table 2; the default MP and the two single-loss MP variants are from the ablation runs. Mean ± std over three seeds.

| | CUB-200 | | Stanford Cars | |
|---|---|---|---|---|
| Method | T1↑ | ECE↓ | T1↑ | ECE↓ |
| KD | 81.75 ±0.02 | 20.67 ±0.26 | 71.88 ±0.58 | 30.01 ±0.32 |
| MP, full ($\mathcal{L}_{\mathrm{mean}} + \mathcal{L}_{\mathrm{proj}}$) | **82.46** ±0.19 | 15.03 ±0.21 | **73.64** ±0.46 | 18.13 ±0.16 |
| MP, $\mathcal{L}_{\mathrm{mean}}$ only | 81.74 ±0.16 | **13.35** ±0.40 | 71.88 ±0.50 | **15.61** ±0.36 |
| MP, $\mathcal{L}_{\mathrm{proj}}$ only | 82.10 ±0.11 | 13.91 ±0.39 | 73.03 ±0.59 | 17.53 ±0.33 |

Table E.12: CIFAR-100 from-scratch distillation under the standard CRD/`mdistiller` protocol. The Single-sample row is the pointwise squared-Hellinger endpoint of MP, obtained when $|S_k| = 1$. Results are mean ± standard deviation over three seeds.

| | WRN-40-2 → WRN-16-2 | | ResNet-32×4 → ResNet-8×4 | | ResNet-32×4 → MobileNet-V2 | |
|---|---|---|---|---|---|---|
| Method | T1↑ | ECE↓ | T1↑ | ECE↓ | T1↑ | ECE↓ |
| CE (no teacher) | 73.62 ± 0.17 | 8.58 ± 0.24 | 72.78 ± 0.27 | 8.15 ± 0.47 | 65.43 ± 0.17 | 12.32 ± 0.10 |
| KD | 75.09 ± 0.11 | 12.39 ± 0.12 | 73.94 ± 0.12 | 11.07 ± 0.04 | 67.33 ± 0.44 | 20.14 ± 0.42 |
| Single-sample (ours, Hellinger) | 74.71 ± 0.09 | 12.33 ± 0.19 | 73.79 ± 0.14 | 11.64 ± 0.26 | 67.15 ± 0.04 | 19.03 ± 0.25 |

Table E.13: DKD $\beta_{\mathrm{dkd}}$ sweep on the Oxford-IIIT Pets fine-grained backbones, complementing Table 5. The default $\beta_{\mathrm{dkd}} = 8$ corresponds to the values in Table 2; $\beta_{\mathrm{dkd}} \in \{2, 4\}$ are additional sweep points. Mean ± std over three seeds.

| | $\beta_{\mathrm{dkd}} = 2$ | | $\beta_{\mathrm{dkd}} = 4$ | | $\beta_{\mathrm{dkd}} = 8$ (default) | |
|---|---|---|---|---|---|---|
| Backbone | T1↑ | ECE↓ | T1↑ | ECE↓ | T1↑ | ECE↓ |
| ViT-B/16 → DeiT-Tiny | 91.62 | **5.65** | 91.64 | 6.27 | 91.56 | 9.63 |
| ViT-B/16 → ViT-Tiny | 91.06 | **9.65** | 91.11 | 10.96 | 90.88 | 15.02 |
| DINOv2-B/14 → DeiT-Tiny | 91.82 | **4.86** | 92.04 | 5.54 | 91.83 | 7.07 |
| DINOv2-B/14 → ViT-Tiny | 90.90 | **9.16** | 91.03 | 10.24 | 90.89 | 13.95 |

