# OpenReview forum: "Relaxing Pointwise Imitation in Knowledge Distillation with Mean-Projection Matching"
_TMLR — Under review for TMLR_

### Review · Reviewer_PSPS · 2026-06-27

**Summary Of Contributions:**

Summary:
This paper proposes Mean Projection (MP), a logit-based knowledge distillation objective that relaxes pointwise imitation of the teacher's predictive distribution. MP maps predictions to the square-root probability sphere, partitions the mini-batch into gold-label subsets, and decomposes each subset's empirical mean vector into a direction and a length. The resulting objective has two bounded terms: mean-direction matching at the subset level and projection matching at the sample level. The paper argues that this preserves useful teacher summaries while allowing the student to deviate from exact pointwise copying, and reports improved calibration and teacher-error inheritance on NLI, fine-grained classification, and domain-shift benchmarks.

Strengths:
- The paper has a clear and well-motivated conceptual contribution. The shift from pointwise targets to subset-level summaries on the square-root sphere is geometrically intuitive, and the admissible-set interpretation gives a principled way to understand what MP preserves and relaxes.
- The per-sample mechanism analysis (recovery, damage, mimicry) is a strong empirical addition. Tracing how often the student recovers teacher-wrong samples versus copies teacher mistakes gives concrete evidence for the claimed relaxation effect.
- The theoretical grounding is solid. The single-sample Hellinger endpoint, the vMF-inspired directional-statistics lens, and the gradient-scaling analysis all support the method's design, and the boundedness of the two loss terms is a practical advantage.

Weaknesses:
- My biggest concern is the breadth of the empirical comparison. The paper positions MP as a general logit-distillation objective, yet the experiments only compare against three baselines: standard KD, DKD, and RLD. Several recent logit-based methods cited in the paper (MLKD, CTKD, LSKD, REDistill) are not evaluated, and the exclusion is not justified. Given that some of these methods also target calibration or teacher imperfection, a broader comparison is necessary to establish MP's practical value.
- The task coverage is narrow relative to the generality of the claim. The main vision experiments are limited to three fine-grained datasets with relatively small class counts (37–200 classes) and lightweight student backbones (5–6M parameters). There are no results on ImageNet, on larger student architectures, or on high-K regimes where class-wise subsets would be sparse. The paper explicitly notes that large-K settings are left to future work, which weakens the claim of general applicability.
- The NLI results do not clearly support the imperfect-teacher narrative. On FEVER-NLI and WANLI, RLD outperforms MP for both student backbones, and on ANLI average MP is only strongest for the smaller ELECTRA-small student. Since the paper uses NLI as a direct diagnostic for imperfect-teacher distillation, the fact that MP is not the best method on the majority of these splits undermines the empirical argument.
- The fine-grained top-1 accuracy claims should be stated more carefully. MP does not consistently win on top-1: on CUB-200 with ViT-B/16 teacher, MP is 77.83% versus RLD's 78.73% for DeiT-Tiny, and on ViT-Tiny it is 82.44% versus RLD's 82.78%. While the calibration advantage is clear, framing MP as matching "the strongest baseline on accuracy" is only true in aggregate and not per-setting.
- The subset-relaxation effect is weak in the main experimental regime. Table 6(b) shows that for CUB-200 (K=200) and Stanford Cars (K=196) with batch size 256, the expected subset size is only ~1.3, and the [MP − single-sample] gap is small or statistically marginal. This means much of the reported benefit comes from the single-sample Hellinger endpoint (i.e., the square-root geometry) rather than from the subset-level relaxation that is the paper's core conceptual contribution.
- Hyperparameter choices across methods are not fully fair. MP uses T=2 while KL baselines use T=4, and MP uses multiplicative mixing (1−α)LCE + αT LMP while DKD/RLD use additive mixing LCE + Lmethod. The DKD β-sweep (Table 5) shows that its published default is often suboptimal on these datasets, which raises the question of whether the other baselines were also given their best chance.
- The Office-Home results are mixed and do not clearly favor MP. Table E.1 shows that MP gives the best OOD−ID NLL gap for DeiT-Tiny and ViT-Tiny, but its OOD ECE on ViT-Tiny (6.31%) is worse than KD (4.49%), DKD (4.22%), and RLD (4.12%), and on DINOv2-S/14 MP is not the best on any reported metric. This inconsistency should be discussed more openly.

**Audience:**

Yes

**Audience Explanation:**

Yes. The paper raises a meaningful question about the supervision object in logit-based distillation, and the geometric perspective (square-root probability sphere, admissible sets, directional statistics) is intellectually interesting. The mechanism analysis (recovery, damage, mimicry) is also a valuable empirical tool that other researchers in knowledge distillation and model compression could adopt. Even if the current empirical scope is limited, the conceptual framework and the analytical methodology could influence future work in this area. TMLR's audience includes researchers interested in theoretical perspectives on learning objectives and in diagnostic analyses of teacher-student behavior, for whom this paper would be relevant.

**Broader Impact Concerns:**

I do not have significant broader impact concerns specific to this work.

**Claims And Evidence:**

No

**Claims Explanation:**

While the paper presents a coherent theoretical framework and some promising empirical signals, I do not believe the claims are fully supported by convincing evidence at this stage. My reasoning follows the weaknesses above.

First, the central claim is that MP improves distillation by relaxing pointwise imitation through subset-level mean-projection matching. However, the mechanism decomposition in Table 6 shows that the single-sample endpoint (pointwise squared Hellinger distance) already accounts for most of the calibration benefit and a substantial portion of the recovery/mimicry improvements. The subset-relaxation component adds only marginal gains in the main experimental regime, where expected subset sizes are close to 1. This weakens the evidentiary link between the core conceptual contribution and the reported empirical gains.

Second, the empirical comparisons are too narrow to support the generality of the claims. The paper positions MP as a broadly applicable logit-distillation objective, yet it is only tested against three baselines on small-scale fine-grained datasets and lightweight students. Missing benchmarks include ImageNet, larger student architectures, and recent competing methods such as MLKD, CTKD, LSKD, and REDistill. Without these comparisons, the claim that MP represents a practically superior distillation objective is unsupported.

Third, the NLI diagnostic, which is meant to demonstrate the imperfect-teacher setting directly, does not clearly favor MP. On two of the three evaluated datasets (FEVER-NLI and WANLI), RLD outperforms MP for both student backbones. This contradicts the narrative that MP is particularly effective when the teacher is imperfect.

Fourth, hyperparameter asymmetries between MP and the baselines (different temperatures, different loss mixing forms) make it difficult to attribute the observed differences to the method itself rather than to tuning choices. The fact that DKD's published default is suboptimal on these datasets (shown by the β-sweep) further suggests that the baselines may not have been given their fairest possible configuration.

Fifth, the Office-Home distribution-shift results are inconsistent: MP underperforms all distillation baselines on OOD ECE for ViT-Tiny and is not the best method for DINOv2-S/14 on any metric. These mixed results are not adequately acknowledged in the main text.

Taken together, the evidence supports the claim that the square-root/Hellinger geometry can improve calibration over standard KL-based KD, but it does not yet convincingly support the stronger claim that subset-level mean-projection matching is a broadly superior distillation paradigm.

**Requested Changes:**

Critical to acceptance:
1. Broader baseline comparison. Include experimental comparisons with at least 2–3 of the recent logit-distillation methods discussed in the related work but not evaluated: MLKD, CTKD, LSKD, or REDistill. If these are truly outside the scope, provide a clear experimental justification (e.g., runtime incompatibility, focus mismatch) rather than silent exclusion.

2. Large-scale validation. Add results on ImageNet-1k (or at least a benchmark with K ≥ 1000 classes and larger students, e.g., ViT-Small or larger). The current experiments are limited to fine-grained datasets with K ≤ 200 and tiny students (5–6M parameters). Without evidence that MP scales beyond this regime, the claim of general applicability is not credible.

3. Fair hyperparameter controls. Report ablations or controls that isolate the effect of the temperature choice: (a) MP with T=4, (b) KD with T=2, under the same mixing form. Alternatively, perform a small grid search over T for all methods on a validation subset. The current asymmetric setting makes it impossible to separate method effects from tuning effects.

Would strengthen the work:
4. Clarify the subset-relaxation contribution. The single-sample Hellinger endpoint explains most of the empirical benefit in the current regime. The paper should either (a) provide experiments with batch sizes large enough to make expected subset sizes clearly larger than 1 on the 200-class datasets, or (b) revise the framing to emphasize that the primary practical benefit currently comes from the square-root/Hellinger geometry, with subset relaxation adding gains mainly when subsets are meaningfully populated.

5. Reconcile the NLI results with the imperfect-teacher narrative. Since MP is not the best method on FEVER-NLI or WANLI, the paper should explain under what conditions (dataset properties, teacher-student capacity gap, task type) MP is expected to outperform label-aware methods like RLD. A failure-mode analysis would strengthen the paper.

6. Present Office-Home results more transparently. Acknowledge explicitly where MP underperforms the baselines (e.g., ViT-Tiny OOD ECE) and discuss possible reasons. Selective reporting of favorable subsets weakens reader trust.

7. Mechanism analysis on NLI. The per-sample recovery/damage/mimicry analysis is currently only reported for fine-grained classification. Adding the same analysis for the NLI setting would strengthen the link between the imperfect-teacher diagnostic and the mechanism claims.

8. ImageNet pretraining consistency. The fine-grained experiments use ImageNet-pretrained teachers and students, which is reasonable, but it introduces a confounder: the benefits may interact with pretraining quality. A brief ablation or discussion of how pretraining strength affects the MP–KD gap would be useful.

---

### Review · Reviewer_CsJh · 2026-07-02

**Summary Of Contributions:**

# Summary
This paper proposes Mean-Projection, a logit-based distillation objective that replaces  exact pointwise imitation of the teacher with matching a class-wise subset mean direction plus a  per-sample scalar projection on the square-root probability sphere, reducing to squared Hellinger distance  in the single-sample limit.

#  Strengths:
  - Solid theoretical backing.
  - Lowest ECE among distillation methods across all 12 fine-grained backbones (avg  13.2% vs 19.6/26.2/19.8 for KD/DKD/RLD) while keeping top-1 on par with the strongest baseline; gains  largest on visually similar classes.
  - MP students recover the gold label on teacher-wrong samples ~2.9pp  more often than KD and copy teacher mistakes ~4.1pp less often, with damage essentially unchanged.

#  Weaknesses:
  - All exp are on small/narrow datasets, not fully convincing. Fine-grained sets, three-class NLI, Office-Home, and one  CIFAR-100 check. no large-scale benchmark (e.g., ImageNet); scaling and the large-K regime are left to  future work.
  - MP is mostly "on par" rather than better on top-1 and is beaten by RLD on several splits (DistilBERT/ANLI, larger DINOv2-S students); the headline benefit is calibration, not accuracy.
  - Relies on non-singleton class-wise subsets, so effectiveness depends on batch size  and K, and the subset-relaxation benefit shrinks as classes become sparse.

**Audience:**

Yes

**Audience Explanation:**

Yes. The knowledge-distillation and model-calibration communities within TMLR's audience would find this  of interest. The reframing of distillation supervision is a genuinely good viewpoint, and the connection to directional statistics (vMF) and the squared-Hellinger endpoint  gives it conceptual depth beyond a routine method paper.

**Broader Impact Concerns:**

no concern

**Claims And Evidence:**

Yes

**Claims Explanation:**

Theory and calibration are well supported; accuracy and generality are not fully convincing.
The "top-1 on par with the best baseline" claim is only partly borne out (RLD wins on several splits; small margins vs. std, no significance testing), and all experiments are on small/narrow datasets with no large-scale benchmark.

**Requested Changes:**

1. would be best if large scale experiments can be added
2.  K_B is overloaded — used as both the set {k : |S_k|>0} and its cardinality
3. Make explicit in the main text that CIFAR-100 tests only the |S_k|=1 Hellinger endpoint, not the full subset-relaxation mechanism

---

### Review · Reviewer_vkQQ · 2026-07-06

**Summary Of Contributions:**

This manuscript proposes Mean Projection (MP), a logit-based knowledge distillation objective designed to mitigate exact pointwise imitation of imperfect teacher predictions. MP maps the teacher and student predictive distributions onto the square-root probability sphere, computes class-wise subset mean directions, and supervises the student through a combination of mean-direction matching and sample-level projection matching. Empirically, the paper evaluates MP across imperfect-teacher NLI, fine-grained image classification, per-sample teacher-error inheritance, distribution shift, and several ablation studies, providing evidence for the effectiveness of the proposed method.

**Audience:**

Yes

**Audience Explanation:**

This manuscript addresses a relevant issue in KD: how to use teacher soft predictions when the teacher is miscalibrated, biased, or partially wrong.

**Claims And Evidence:**

No

**Claims Explanation:**

The main claims and the proposed method are intuitive and are largely well supported by a diverse set of experiments. Specifically, MP consistently reduces ECE compared with the baselines while keeping top-1 accuracy close to the strongest baseline. The per-sample mechanism analysis is also useful and supports the interpretation that MP reduces teacher-error inheritance: recovery increases and mimicry decreases relative to KD, while damage remains nearly unchanged.

However, I have several concerns.

1. The motivation for using the coordinate-wise square-root transformation in Eq. 3 is not sufficiently justified. If the goal is to use summarized information from teacher predictions as useful supervision, one could also consider various alternative transformations, such as the identity or square transformation. However, the manuscript only discusses the square-root transformation and does not provide either theoretical or empirical comparisons against other possible transformations.

2. While the proposed method is simple and intuitive, the paper mostly supports its advantage over prior methods through empirical performance, without providing a theoretical explanation for why MP should outperform existing approaches. The manuscript appears to provide a theoretical interpretation of the proposed method itself, but not a theoretical comparison with prior work.

3. KD is typically framed as transferring knowledge from a high-performing teacher to a student, and I am concerned about the practical realism of the Section 4.1 experiments that use a substantially underperforming imperfect teacher. In practice, if the teacher model performs poorly on the dataset to be distilled, one would likely avoid performing KD with that teacher or simply use a different teacher model.

**Requested Changes:**

I believe that addressing the following points, in addition to the concerns raised above, would further strengthen the study.

1. This study may avoid suggesting that MP is broadly superior to existing distillation methods. The results more precisely support the claim that MP improves calibration and reduces teacher-error mimicry relative to pointwise KD, while remaining competitive in accuracy.

2. Table 6 suggests that much of the ECE improvement comes from the single-sample Hellinger endpoint, whereas subset relaxation contributes more clearly to recovery and mimicry. This work might reflect this decomposition in the main claims, not only in the analysis section.

3. Since MP relies on class-wise mini-batch subsets, its subset-level advantage appears dependent on expected subset size. It might be better if this manuscript more clearly discuss the implications for large-K tasks, small batch regimes, class imbalance, and distributed training.